# 1 Characteristics of snowpack chemistry on the coastal region in the

# 2 northwestern Greenland Ice Sheet facing the North Water

- 3 Yutaka Kurosaki<sup>1</sup>, Sumito Matoba<sup>1</sup>, Mai Matsumoto<sup>1,2</sup>, Tetsuhide Yamasaki<sup>3</sup>, Ilannguaq Hendriksen<sup>4</sup>,
- 4 Yoshinori Iizuka<sup>1</sup>
- <sup>5</sup> Institute of Low Temperature Science, Hokkaido University, Sapporo, 060-0819, Japan
- 6 <sup>2</sup>Graduate School of Environmental Science, Hokkaido University, Sapporo, 060-0810, Japan
- <sup>3</sup>Avangnaq, Takatsuki 596-0094, Japan
- 8 <sup>4</sup>Siorapaluk, Avannaata Kommune, Greenland
- 9 Correspondence to: Yutaka Kurosaki (yutaka kurosaki@lowtem.hokudai.ac.jp),
- 10 Sumito Matoba (matoba@lowtem.hokudai.ac.jp)

#### 11 Abstract

12 In the North Water, the opening of sea ice due to polynya formation may influences the surrounding water and aerosol 13 circulation. We conducted glaciological observations from seacoast to inland on the western side of Prudhoe Land, which is 14 located in the northwestern Greenland Ice Sheet close to the North Water, to elucidate water and aerosol circulation around the North Water. The spatial variations in  $\delta^{18}$ O and chemical substances in surface snow showed that water vapor and aerosols 15 were directly transported from the southern North Water to northern inland of areas on the western side of Prudhoe Land. 16 17 Unlike the inland area of the Greenland Ice Sheet, the snowpack on the western side of Prudhoe Land contained signals of 18 ocean biological and sea ice conditions in the North Water. The spring-summer snowpack contained high concentrations of 19 methanesulfonic acid, which likely derived from sea-to-air dimethylsulfide emission following a phytoplankton bloom. The 20 autumn-winter snowpack exhibited high concentrations of NH<sub>4</sub><sup>+</sup>, which could potentially have originated from sea-to-air ammonia gas emissions during the decline and death phases of marine organisms. The Na<sup>+</sup>, Cl<sup>-</sup>, K<sup>+</sup>, Mg<sup>2+</sup>, SO<sub>4</sub><sup>2-</sup>, and Ca<sup>2+</sup> 21 concentrations in the snowpack rapidly increased in winter, which could be attributed to the emission of frost flowers from the 22 23 newly formed sea ice surface into the atmosphere due to intense storm activity in the North Water. We suggest that the chemical 24 substances identified in the snowpack or ice core from the western side of Prudhoe Land can be used to better understand past 25 changes in ocean biological and sea ice conditions in the North Water.

## 1 Introduction

The North Water (NOW) is one of the largest polynyas in the Arctic and is located between northwestern Greenland and Ellesmere Island in northern Baffin Bay (Barber and Massom, 2007) (Fig. 1a). The polynya in the NOW is formed by strong northerly winds from the Nares Strait and the supply of relatively warm Atlantic waters yielded by a branch of the West Greenland Current (Ito. 1982; Steffen and Ohmura, 1985; Mysak and Huang, 1992; Melling et al., 2001; Ingram et al., 2002; Dumont et al., 2009; Vincent, 2019). Polynya formation in the NOW supplies heat and moisture to the atmosphere and influences low-cloud formation around the NOW region. Monroe et al. (2021) reported that the low-cloud amount over polynyas can reach up to twice as high as that over nearby sea ice in the NOW. Moreover, polynya formation in the NOW influences ocean conditions such as vertical mixing by wind, the advection of nutrients into the euphotic layer, light availability, and density stratification. These physical factors enhance primary production and collectively support high levels of productivity in the NOW (Mei et al., 2002; Marchese et al., 2017). Additionally, the notable phytoplankton blooms promote the production of dimethylsulfide (DMS), and sea-to-air DMS emission contributes to marine cloud formation (e.g. Charlson et al., 1987). Sea ice fluctuations in the NOW influence not only water and aerosol circulation and ocean productivity but also human activities. The recent increase in the frequency of sea ice breakup in front of Siorapaluk village, which is the northernmost village in Greenland, influences hunting and fishing on sea ice and movement between villages on sea ice by dog sledges and skidoos in winter and spring (Matoba and Yamasaki, 2018). Decreases in the sea ice thickness and concentration in the NOW due to future climate warming could result in changes in water and aerosol circulation, ocean productivity, and human lifestyle conditions. To elucidate the changes in the source environmental conditions and the transport process of water vapor and aerosols over northwestern Greenland, snow pit and ice core observations have been conducted in the northwestern Greenland Ice Sheet (Kuramoto et al., 2011; NEEM Community members, 2013; Matoba et al., 2015; Matoba et al., 2018; Osterberg et al., 2015; 

To elucidate the changes in the source environmental conditions and the transport process of water vapor and aerosols over northwestern Greenland, snow pit and ice core observations have been conducted in the northwestern Greenland Ice Sheet (Kuramoto et al., 2011; NEEM Community members, 2013; Matoba et al., 2015; Matoba et al., 2018; Osterberg et al., 2015; Kurosaki et al., 2020). The chemical components and water stable isotopes in the snowpack on the Greenland Ice Sheet provide valuable information related to past changes in aerosol and water vapor conditions in source regions and their transport processes. Snow pit observations (NEEM; 77.45°N, 51.06°W, 2445 m above sea level (a.s.l.) (Fig. 1a)) at the inland of the northwestern Greenland Ice Sheet revealed seasonal variations in snowpack chemical compositions (Kuramoto et al., 2011). For example, the concentration of Ca<sup>2+</sup> originating from mineral dust peaked from late winter to spring, the concentrations of SO<sub>4</sub><sup>2-</sup> and NO<sub>3</sub><sup>-</sup> originating from fossil fuel combustion peaked from winter to spring, and the concentration of methane sulfonate (hereafter referred to as MSA) originating from marine phytoplankton peaked in spring and from late summer to autumn (Kuramoto et al., 2011). Analysis of mineral dust in an ice core from SIGMA-D (77.636°N, 59.120°W, 2100 m a.s.l.), which is located approximately 250 km from Qaanaaq on the northwestern coast of Greenland (Fig. 1a), revealed that the amount of local dust originating from the Greenland coastal region increased recently due to a decrease in snow cover during the warm season (Nagatsuka et al., 2021). Kurosaki et al., (2020) reported that the temporal variation in deuterium excess (dexcess) in an ice core from SIGMA-A (78.05°N, 67.63°W, 1490 m a.s.l.), which is located approximately 70 km northeast of

Qaanaaq (Fig. 1a and b), was significantly correlated with the sea ice concentration in northern Baffin Bay from 1979–2015. Via the use of relationship from the SIGMA-A ice core, the temporal variation in sea ice concentrations over 100 years in northern Baffin Bay was reconstructed (Kurosaki et al., 2020). Because all of these studies were performed at relatively high altitudes and far from the coast, the variations in chemical compositions and water stable isotopes were reflected by environmental changes in relatively large areas driven by large-scale atmospheric circulation rather than local phenomena. Matoba et al. (2014) conducted multiple snow pit observations at sites closer to the coast on the eastern side of Prudhoe Land, which is located the northern part of Hayes peninsula in the northwestern Greenland facing the northeastern side of the NOW (Fig. 1a and b), and they reported notable spatial variations in water stable isotopes and chemical compositions at lower elevations and coastal regions. Water vapor and sea salt on the glacier facing the ocean were transported from the coast. On the other hand, water vapor, mineral dust, anthropogenic substances, and MSA on the eastern side of Prudhoe Land were transported from the central-west coast of Greenland, which is located around Disko Bay, via the central part of the Greenland Ice Sheet (Matoba et al., 2014). The changes in the NOW could influence the water and aerosol circulation in the northwestern Greenland. However, the past changes in the NOW have never been traced from the previous snow and ice core observations in the northwestern Greenland Ice Sheet.

The western side of Prudhoe Land facing the northeastern side of the NOW is located between Inglefield Land in the north and the southern part of Hayes peninsula in the south (Fig. 1a and b). The Inglefield Land is dry area where the precipitation amount is quite low. On the other hand, the southern part of Hayes peninsula is one of the highest areas of precipitation amount (> 600 mm yr<sup>-1</sup>) in the northwestern Greenland due to the southerly moisture advection (Bales et al., 2001). The western side of Prudhoe Land, where locates between these regions with large contrast in precipitation, could be influenced by complicate processes of the heat and moisture supply from the ocean around this region including the NOW. However, most of meteorological and glaciological observations have been conducted on the southern part of Hayes peninsula in the northwestern Greenland (Bales et al., 2001; Osterberg et al., 2015; Becagli et al., 2016; Akers et al., 2020), not on Prudhoe Land. In order to make accurate future projections of environmental change in this region, where the influences of climate warming on the life activities in this region is already emerging, it is necessary to clarify the impact of changes in the NOW on the processes of the heat and moisture circulation in this region.

In the Prudhoe Land, a gentle valley separates the western region from the eastern region, where the SIGMA-A site is situated. (Fig. 1b). This valley could serve as a pathway for air masses transported from inland of the Greenland Ice Sheet to descend toward the coastal region, and these air masses are likely not transported to the western side of Prudhoe Land. If that is the care, the snowpack on the western part of Prudhoe Land could contain aerosols originating from the NOW without being mixed with aerosols from the interior of the Greenland Ice Sheet. We conducted glaciological observations on the western side of Prudhoe Land facing the NOW to elucidate water and aerosol circulation around the NOW, which influences the environmental changes and related human activities in this region.

**Figure 1: Maps of the sampling sites.** (a) shows location of the snowpit and ice core sampling sites in this study (St. 9) and previous studies (SIGMA-A, SIGMA-D, and NEEM) in the northwestern Greenland Ice Sheet. The dashed polygon in (a) denotes the approximate location of the NOW. Hayes peninsula in the northwestern Greenland is located between Kane Basin in the north and Melville Bay in the south. (b) shows Landsat-8 image around St. 9 and SIGMA-A of Prudhoe Land, which is located on the northern part of Hayes peninsula, on 13 April 2023. The black circles in (b) denote the sampling sites from St. 1 to St. 9, and the black line denotes dog sledge route. The gray contours in (b) are drawn from the Greenland Mapping Project 2 (GIMP-2) Digital Elevation Model version 2.

## 2 Methods

99

100

#### 2.1 Observations and chemical analysis

We conducted snow observations from 9-11 April 2023 on the western side of Prudhoe Land, which is a coastal site in the 102 northwestern part of the Greenland Ice Sheet (Fig. 1). The expedition base was established in Siorapaluk village at 77.78°N 103 and 70.75°W. Siolapaluk village faces Robertson Fjord, which is located on the northeastern side of the NOW (Fig. 1b). The 104 observation route is shown in Fig. 1b. We started from Siorapaluk on 9 April 2023 on dog sledges, moved on sea ice westward 105 from Sioraparuk, climbed the Clements Glacier, and advanced on the Greenland Ice Sheet. We arrived at the terminal point of 106 the route (78.13°N, 71.06°W, 1279 m above sea level (St. 9)) on 10 April 2023 and returned to Siorapaluk on 11 April 2023. 107 We conducted glaciological observations along the route. The locations and elevations of the observation sites are shown in 108 Fig. 1b and Fig. S1. We conducted snow pit observations on the glacier (St. 3) and the ice sheet (St. 9) along the route to 109 measure the chemical species, stratigraphy, and density of the snowpack. Snow samples for chemical analysis at St. 3 and St. 110 9 were collected from the snow walls of the snow pit using a precleaned stainless-steel sampler into <sup>®</sup>Whirl-Pak polyethylene 111 bags (Nasco). The snow sampling intervals at St. 3 were 0.02 m from 0.00 to 0.20 m and 0.03 m from 0.20 to 1.01 m, and the 112 snow sampling intervals at St. 9 were 0.02 m from 0.00 to 0.30 m and 0.03 m from 0.30 to 1.08 m. Additionally, an ice core 113 was collected at St. 9 from 1.08 to 4.20 m using a hand corer. After the stratigraphy and density of the ice core were measured, the ice core samples were cut at 0.05-0.10 m intervals using a precleaned saw, were polished using a precleaned ceramic knife 114 to remove contamination on the ice core samples and were then placed into ®Whirl-Pak polyethylene bags (Nasco). Surface 115 116 snow samples for chemical analysis were also collected at 9 sites along the route using a precleaned stainless-steel sampler 117 into Whirl-Pak polyethylene bags. We removed the potential contamination by organic compounds in general on the material and tools used for contamination removal using ethanol, and then performed ultrasonic cleaning in ultrapure water. The snow 118 119 samples were kept frozen and were transported to Siorapaluk. The snow samples were melted at ambient temperature, were 120 placed in 50 mL clean polypropylene bottles, and were then kept frozen in a portable freezer (approximately -18 °C). The 121 samples were then transported to a cold room (-20 °C) at the Institute of Low Temperature Science (ILTS) of Hokkaido 122 University, Japan, and were then kept frozen until chemical analysis. The concentrations of Na<sup>+</sup>, NH<sub>4</sub><sup>+</sup>, K<sup>+</sup>, Mg<sup>2+</sup>, Ca<sup>2+</sup>, MSA, Cl<sup>-</sup>, SO<sub>4</sub><sup>2-</sup>, and NO<sub>3</sub><sup>-</sup> were measured by ion chromatography (ICS-123 2100, Thermo Scientific). For the cations, separation was carried out with a Dionex CG-12 (4 × 50 mm) guard column, 124 125 followed by a Dionex CS12-A (4 × 250 mm) separation column. Injection volume of samples was 500 μL. MSA (20 mM) was 126 used as eluent, and flow-rate was kept 1.0 mL min<sup>-1</sup>. Dionex CDRS600 dynamically regenerated suppressor was used for 127 conductivity suppression before conductivity cell. For the anions, separation was obtained with a Dionex AG-18 (4 × 50 mm) 128 guard column and Dionex AS-18 (4 × 250 mm) separation column. Injection volume of samples was 1000 µL. KOH (23 mM) 129 was used as eluent, and flow-rate was kept 1.0 mL min<sup>-1</sup>. Dionex ADRS 600 dynamically regenerated suppressor was used for 130 conductivity suppression before conductivity cell. The absolute calibration curve method was used for quantitative determination of each ion concentration. For the absolute calibration determination by ion chromatography, we used the standard solutions for ion chromatography produced by FUJIFILM Wako Pure Chemical corporation, diluted to 20, 50, 100. and 200 ppb with ultra-pure water. If the ion concentration of samples were outside the calibration range (> 200 ppb), it was remeasured using 500, 1000, 2000-3000, and 6000 ppb standard for the anions and 500, 1000, 2000, and 4000 ppb standard for the cations. Blanks were always evaluated before the calibration procedure. The analytical precision of the ion chromatography was < 5 % (at the measurement of 20 ppb standard). The limit of detection (LOD) was < 0.1 ppb. The limit of quantification (LOO) was < 0.5 ppb. The ion chromatography method in this study had been evaluated by our previous study (Kurosaki et al., 2020; Kurosaki et al., 2022).

The non-sea-salt (nss) components of  $Ca^{2+}$  and  $SO_4^{2-}$  were calculated using the seawater ratios of these ions with respect to Na<sup>+</sup>, thereby assuming that Na<sup>+</sup> is solely of a sea-salt origin. For example, the concentration of nssSO<sub>4</sub><sup>2-</sup> was calculated following Eq. (1):

$$[nssSO_4^{2-}] = [SO_4^{2-}] - (SO_4^{2-}/Na^+)_{sea} \times [Na^+], \tag{1}$$

We applied the concentration unit as μeq L<sup>-1</sup> in the Eq: (1). The (SO<sub>4</sub><sup>2-</sup>/Na<sup>+</sup>)<sub>sea</sub> is the equivalent concentration ratio of SO<sub>4</sub><sup>2-</sup>/Na<sup>+</sup> in the sea water, which is 0.12. Similarly, the concentrations of nssCa<sup>2+</sup> were calculated using sea water ratios of 0.044 (Wilson, 1975; Legrand and Mayewski, 1997).

The stable oxygen and hydrogen isotopic compositions of the water samples were measured by a water stable isotopes analyzer (L2130-I, Picarro Inc.) with an evaporating device (A0211, Picarro Inc.). We used the ultrapure water ( $\delta^{18}O = -11.583$  and  $\delta D = -77.2$ ), Antarctic iceberg ( $\delta^{18}O = -20.4$  and  $\delta D = -158.7$ ), snowpack on the Antarctic ice Sheet ( $\delta^{18}O = -46.694$  and  $\delta D = -370.7$ ) for calibration. The analysis precisions for  $\delta^{18}O$  and  $\delta D$  were 0.08% and 0.8%, respectively.

#### 2.2 Meteorological and sea ice data

We used meteorological and sea ice data from northwestern Greenland and Baffin Bay to analyze the factors contributing to the variations in chemical species and water stable isotopes in snow on the northwestern coast of Greenland. The temperature, relative humidity, wind speed, wind direction, and atmospheric pressure at Siorapaluk were measured by an automatic weather station (AWS) (WXT530, Vaisala) from April 2021 to March 2024 (Matoba et al., 2024). The AWS was installed 2 m above ground level (a.g.l.), and these meteorological factors were observed at 10-minutes intervals. We used data of the air temperature, geopotential height, wind speed, and wind direction around the northwestern coast of Greenland and Baffin Bay from ERA5 reanalysis dataset supplied by the European Center for Medium Range Weather Forecasts (ECMWF) (Hersbach et al., 2020). Additionally, we used 2 m air temperature in the western side of the Prudhoe Land from ERA5-Land reanalysis dataset supplied by the ECMWF (Muñoz-Sabater et al., 2021).

## 2.3 Backward trajectory

165

166 167

172

188

To investigate the source region and transport pathway of water vapor and aerosols contained in ice core at the St. 9 site, we analyzed air mass position along the backward trajectory from the St. 9 site during the past 7 days using the National Oceanographic and Atmospheric Administration (NOAA) Hybrid Single-Particle Lagrangian Integrated Trajectory (HYSPLIT) model (Stein et al., 2015) and National Centers for Environmental Prediction (NCEP) reanalysis data. The initial positions of air mass were set at 50, 500, 1000, 1500 m above ground level over the St. 9 site. The initial date and time were every 6 h from 2019 to 2023. We calculated existing probability of an air mass with a 1° resolution. The existence probability was defined as the proportion of backward trajectories originating from the St. 9 that passed through each 1°×1°grid cell. We counted the number of times the backward trajectories originating from St. 9 passed through each grid cell. Subsequently, we normalized the count for each grid cell by dividing it by the total number of the trajectory passes across all grid cells. Considering the water vapor and aerosols supply from the ocean and land surface, we excluded air mass over 1000 m above ground level. The existence probability was weighted by the daily amount of precipitation when the air mass arrived at the St. 9 site. The daily amount of precipitation was extracted from the ERA5 reanalysis dataset (Hersbach et al., 2020).

#### 3 Results and Discussion

## 3.1 Snowpack dating and annual accumulation

The snowpack at St. 3 and St. 9 was dated on the basis of the snow stratigraphy and seasonal variations in  $\delta^{18}$ O, d-excess (d-excess =  $\delta D - 8\delta^{18}O$ ) and MSA. The vertical profiles of the snow stratigraphy at St. 3 and St. 9 are shown in Fig. S2. Rounded grains, faceted crystals, and depth hoars were observed from 0.00 to 1.01 m in the glacier at St. 3 (Fig. S2), Glacier ice was present below 1.01 m at St. 3. The snowpack from 0.00 to 1.01 m at St. 3 corresponds to seasonal snow for the periods from autumn in 2022 to spring in 2023. Rounded grains, faceted crystals, and depth hoars were observed from 0.00 to 0.96 m at St. 9 (Fig. S2). Below 0.96 m at St. 9, melt forms prevailed, and some ice layers with thickness ranging from several millimeters to 10 cm were found (Fig. S2). The ice layers indicated that the surface or subsurface of the snowpack melted in summer. Therefore, we assumed that the depth of 0.96 m at St. 9 corresponds to the end of summer in 2022. Vertical profiles of  $\delta^{18}$ O, d-excess and MSA at St. 9 are shown in Fig. 2. The seasonal variation in  $\delta^{18}$ O reflects the air temperature and exhibits a maximum value in summer and a minimum value in winter at several Greenland sites (Steffensen et al., 1988; Legrand and Mayewski, 1997; Kuramoto et al., 2011; Kurosaki et al., 2020; Nakawaza et al., 2021). The seasonal variation in d-excess reflects the sea surface temperature and sea-to-air evaporation process in the water vapor source region of precipitation (Merlivat and Jouzel, 1979; Uemura et al., 2007; Kurita, 2011; Kopec et al., 2019). At several sites in Greenland, the d-excess of snowpack exhibits maximum values in autumn and minimum values from spring to early summer (Johnsen et al., 1989; Kuramoto et al., 2011). MSA is an oxidization product from DMS, which originates from ocean phytoplankton. The seasonal variation in MSA exhibits maximum values from spring to summer during ocean phytoplankton blooms and minimum values in winter (Jaffrezo et al., 1994; Legrand and Mayewski, 1997; Kuramoto et al., 2011; Osterberg et al., 2015; Nakazawa et al., 2021; Kurosaki et al., 2022). We determined the seasonality of the snowpack from 0.00 to 1.15 m as shown below. The date of snow layer from 0.00-0.04 m was close to the observation date (April 2023). We determined that snowpack in 0.04-0.72 m corresponded to the autumn-winter period of 2022–2023, based on the presence of negative  $\delta^{18}$ O peak, positive d-excess peak and low MSA concentrations (Fig. 2). The snowpack from 0.72 m-1.15 m was interpreted to correspond to the springsummer period in 2022, due to the presence of ice laver, high  $\delta^{18}$ O value and high MSA concentrations (Fig. 2 and Fig. S2). Snowpack below 0.96 m corresponded to previous summer and before it because the amplitude of seasonal variation of  $\delta^{18}$ O and d-excess below 0.96 m were smaller than those in the shallower layers from 0.00–0.96 m because of summer melting (Fig. 2a and b). The MSA concentration showed obvious seasonal variations, and the  $\delta^{18}$ O values exhibited slight seasonal variations below 0.96 m (Fig. 2a and c), although the values of the water stable isotopes were smoothed by summer melting and some chemical species showed high peaks in the ice layers owing to relocation processes by meltwater refrozen. We assigned the snow layers with high MSA and  $\delta^{18}$ O values to the spring to summer period and assigned the snow layers with low MSA and  $\delta^{18}$ O values to autumn to winter period. Consequently, the firn layers from 1.15–1.45 m, 2.02–2.28 m, and 3.12–3.39 m corresponded to the autumn-winter periods from 2021-22, 2020-21, and 2019-20, respectively. The snowpacks from 1.45-2.02 m and 2.28-3.12 m corresponded to the spring-summer periods in 2021 and 2020, respectively. According to the snowpack dating approach described above, the snow densities, which were the average of the bulk density of ice core at St. 9 (Fig. S3), from autumn-winter 2020 to autumn-winter 2021 and from autumn-winter 2019 to autumn-winter 2020 were 477 kg m<sup>-3</sup> and 497 kg m<sup>-3</sup>, respectively. Using these mean densities, the annual accumulations were calculated to be 0.41 m w.eq. vr<sup>-1</sup> and 0.56 m w.eq. yr<sup>-1</sup>, respectively. The Fifth Generation Mesoscale Model modified for polar climates (Polar MM5) using firn core and meteorological station data estimated that the climatological value of the annual accumulation rate on the inland of the western side of Prudhoe Land ranged from 0.31-0.43 m w.eg. vr<sup>-1</sup> for the 1958-2007 period (Burgess, et al., 2007). The annual accumulation rate derived from the snowpack at St. 9 in this study was comparable to or slightly higher than the climatological value.

197

202203

206207

208209

213214

Figure 2. Vertical profiles of (a)  $\delta^{18}$ O, (b) d-excess, and (c) MSA at St. 9. The LOD of MSA was < 0.0011  $\mu$ eq L<sup>-1</sup>.

## 3.2 Spatial and temporal variations in $\delta^{18}$ O and chemical species

## 3.2.1 $\delta^{18}$ O

The spatial variations in  $\delta^{18}$ O in the surface snow showed maximum and minimum values at St. 3 (-19.12 ‰) and St. 9 (-37.21 ‰), respectively (Fig. S4a). The average  $\delta^{18}$ O value from 0.00 to 1.01 m at St. 3 was greater than that at St. 9 (St. 3: -22.03 ‰; St. 9: -29.12 ‰) (Table 1). The  $\delta^{18}$ O values in surface snow and the snowpack decreased from the seacoast toward the inland site. The past 7 days backward trajectory arriving at the St. 9 also exhibited that majority of air mass was transported from the south of the St. 9, situated on the northern Baffin Bay and eastern NOW (Fig. S5). We suggest that the south-to-north gradient of  $\delta^{18}$ O results from water vapor, which originates from northern Baffin Bay and eastern NOW, transport from the southern coast to the northern inland area by southerly winds. On the other hand, Matoba et al. (2014) reported different spatial variations in  $\delta^{18}$ O in surface snow on the eastern side of Prudhoe Land. The  $\delta^{18}$ O value of surface snow on the northwestern coast of Greenland decreased along the elevation on a glacier from sea level to 1,100 m a.s.l. and increased toward the inland on the Greenland Ice Sheet. Matoba et al. (2014) suggested that the distribution of  $\delta^{18}$ O values in surface

snow resulted from the water vapor pathway, which originates from southern Baffin Bay, after which it first progress toward the northeast and then moves the inland of the Greenland Ice Sheet, and it finally turns northwest on the western coast of Greenland because of the cyclonic flow into a depression at Melville Bay. Because the observation site in this study is located in the westernmost part of Prudhoe Land, it is not affected by water vapor passing through the interior of the Greenland Ice Sheet, and water vapor transport from the south mainly influences the spatial variation in  $\delta^{18}$ O.

238239

243244

249250

255256

258259260261

Fig. 3 shows the vertical profile of  $\delta^{18}$ O in the snowpack at St. 3 and St. 9, and the difference in  $\delta^{18}$ O between St. 9 and St. 3. The vertical profile of  $\delta^{18}$ O in the snowpacks at St. 9 was similar to that at St. 3. The depths of the negative and positive peaks of  $\delta^{18}$ O at St. 9 agreed well with those at St. 3, and the vertical profile of  $\delta^{18}$ O between 0.00 and 1.01 m at St. 9 correlated significantly with that at St. 3 (r = 0.69, p < 0.01). The snowpack corresponding to autumn–winter from 2022–2023 at St. 3 and St. 9 at the same snow depth were most likely accumulated with precipitation attributed to the same snowfall events, and  $\delta^{18}$ O in the snowpack had not been changed by the post depositional processes, which is melting and refreezing, water molecule diffusion, wind blowing, and sublimation. Therefore, we propose that the vertical profile of  $\delta^{18}$ O between 0.00 and 1.01 m at St. 9 can be reasonably compared with the profile at St. 3 based on their differences. The difference in  $\delta^{18}$ O values between St. 3 and St. 9 increased from spring-summer to autumn-winter from 2022-2023 and decreased until spring in 2023 (Fig. 3c). Matoba et al. (2002a) reported that the  $\delta^{18}$ O values of precipitation in Siorapaluk was positively correlated with the surface air temperature. We suggest that the altitude gradient of the surface air temperature in winter was greater than that in summer in the western region of Prudhoe Land. We estimated the difference in surface air temperature between St. 3 and St. 9 using ERA5-Land reanalysis dataset (Fig. S6 and Fig. S7). The temperature difference between St. 3 and St. 9 was smallest in summer and increased toward winter. The mean temperature differences in autumn and winter were larger than those in summer. This result supports our suggestion, based on water stable isotope, that the altitude gradient of surface air temperature in the western side of Prudhoe Land was steeper in winter than in summer.

Table 1. Mean values and standard deviations of several ion species and water stable isotopes between St. 3 and St. 9. The mean values at St.9 were obtained at depths from 0.00 to 1.01 m.

|             | Na <sup>+</sup><br>(µeq L <sup>-1</sup> ) | NH <sub>4</sub> +<br>(µeq L-1) | Κ <sup>+</sup><br>(μeq L <sup>-1</sup> ) | Mg <sup>2+</sup><br>(µeq L <sup>-1</sup> ) | nssCa <sup>2+</sup><br>(µeq L <sup>-1</sup> ) | MSA<br>(µeq L <sup>-1</sup> ) | CI <sup>-</sup><br>(µeq L <sup>-1</sup> ) | nssSO <sub>4</sub> <sup>2-</sup><br>(µeq L <sup>-1</sup> ) | NO <sub>3</sub> -<br>(μeq L <sup>-1</sup> ) | Cl <sup>-</sup> /Na <sup>+</sup><br>(µeq L <sup>-1</sup> ) | δ <sup>18</sup> Ο<br>(‰) | δD<br>(‰) | d-excess<br>(‰) |
|-------------|-------------------------------------------|--------------------------------|------------------------------------------|--------------------------------------------|-----------------------------------------------|-------------------------------|-------------------------------------------|------------------------------------------------------------|---------------------------------------------|------------------------------------------------------------|--------------------------|-----------|-----------------|
| mean(St.3)  | 133.70                                    | 0.61                           | 2.77                                     | 22.69                                      | -0.46                                         | 0.01                          | 155.53                                    | 1.51                                                       | 1.32                                        | 1.19                                                       | -22.03                   | -160.27   | 15.93           |
| std. (St.3) | 181.51                                    | 0.45                           | 4.26                                     | 27.51                                      | 0.87                                          | 0.02                          | 206.34                                    | 3.45                                                       | 1.19                                        | 0.15                                                       | 2.22                     | 17.58     | 2.12            |
| mean(St.9)  | 34.17                                     | 0.04                           | 0.68                                     | 7.11                                       | 0.85                                          | 0.01                          | 41.37                                     | 0.40                                                       | 0.60                                        | 1.17                                                       | -29.12                   | -218.69   | 14.29           |
| std. (St.9) | 54.12                                     | 0.22                           | 1.25                                     | 9.12                                       | 3.67                                          | 0.01                          | 65.33                                     | 0.92                                                       | 0.44                                        | 0.12                                                       | 6.46                     | 52.27     | 2.66            |

Figure 3: Vertical profiles of  $\delta^{18}$ O. (a) and (b) show  $\delta^{18}$ O values at St. 3 and St.9, respectively. (c) shows difference between St.3 and St.9 in terms of  $\delta^{18}$ O.

## 3.2.2 Na<sup>+</sup> and Cl<sup>-</sup>

The possible sources of Na<sup>+</sup> and Cl<sup>-</sup> are sea salt and frost flowers (Legrand and Mayewski, 1997). The Na<sup>+</sup> and Cl<sup>-</sup> concentrations in surface snow decreased with increasing distance from the seacoast (Fig. S4b and g). The average values of Na<sup>+</sup> and Cl<sup>-</sup> between 0.00 and 1.01 m at St. 3 were four times greater than those at St. 9 (Table 1). High values of these ions at St. 3 and St. 9 were found from autumn–winter (Fig. 4). The spatial variations in Na<sup>+</sup> and Cl<sup>-</sup> in surface snow were similar to those in the eastern region of Prudhoe Land (Matoba et al., 2014). The Na<sup>+</sup> and Cl<sup>-</sup> concentrations at several sites in the inland of the Greenland Ice Sheet also exhibit maximum values in winter (Steffensen, 1988; Fischer and Wagenbach, 1996; Dibb et al., 2007; Kuramoto et al., 2011).

The Na<sup>+</sup> and Cl<sup>-</sup> concentrations in snow are modified with respect to the bulk seawater composition due to heterogeneous reactions with atmospheric acids such as H<sub>2</sub>SO<sub>4</sub> and HNO<sub>3</sub> (Cadle, 1972; Delmas et al., 1982; Kerminen et al., 2000):

$$277 \quad 2\text{NaCl} + \text{H}_2\text{SO}_4 \rightarrow \text{Na}_2\text{SO}_4 + 2\text{HCl}, \tag{R1}$$

$$278 \quad \text{NaCl} + \text{HNO}_3 \rightarrow \text{NaNO}_3 + \text{HCl}, \tag{R2}$$

Na<sub>2</sub>SO<sub>4</sub> and NaNO<sub>3</sub> are removed from the atmosphere through the rainout process following the formation of cloud condensation nuclei and precipitation. Therefore, the difference of the Cl<sup>-</sup>/Na<sup>+</sup> ratio in snowpack from seawater ratio often reveals sea salt modification within the atmosphere. In general, if sea salt is transported for a long time in the atmosphere, Na<sup>+</sup> is scavenged to a greater degree than Cl<sup>-</sup> due to reactions with H<sub>2</sub>SO<sub>4</sub> and HNO<sub>3</sub>, and Cl<sup>-</sup>/Na<sup>+</sup> is thus higher than the seawater ratio (1.17). The Cl<sup>-</sup>/Na<sup>+</sup> ratio in the snowpack at St. 9 was close to the seawater ratio for all seasons (Fig. 4). Additionally, the values of Cl<sup>-</sup>/Na<sup>+</sup> in the snowpack at St. 3 and in surface snow at all the sites were close to the seawater ratio (Figs. S3k and 4). At several sites in the northwestern Greenland Ice Sheet, Cl<sup>-</sup>/Na<sup>+</sup> exhibited maximum values in summer (Kuramoto et al., 2011; Kurosaki et al., 2020). In contrast, the Cl<sup>-</sup>/Na<sup>+</sup> ratio at St.9 did not show clear seasonality. Moreover, the spatial distribution of air mass transport to the St.9 exhibited that primary source of sea salt was the area near the St. 9, situated on the eastern NOW, throughout the year (Fig. S5b). Therefore, the sea salt observed in this study could be transported along a short-distance pathway without reactions with H<sub>2</sub>SO<sub>4</sub> and HNO<sub>3</sub> in the atmosphere throughout the year, and the possible source region is the NOW polynya.

Figure 4: Vertical profiles of Na<sup>+</sup>, Cl<sup>-</sup>, and Cl<sup>-</sup>/Na<sup>+</sup> at (a–c) St. 3 and (d–f) St. 9. Green lines denote mean ion concentration across all observation depths. Orange and brown lines denote the mean ion concentrations plus and minus one standard deviation across all observation depths, respectively. The gray lines in (c) and (f) denote Cl<sup>-</sup>/Na<sup>+</sup> in seawater (1.17). The LOD of Na<sup>+</sup> and Cl<sup>-</sup> was < 0.0043 and < 0.0028  $\mu$ eq L<sup>-1</sup>, respectively.

300301

303304

311312

#### 3.2.3 NH<sub>4</sub><sup>+</sup>

The possible source of NH<sub>4</sub><sup>+</sup> is biomass burning (Fuhrer and Legrand, 1997; Legrand and Mayewski, 1997; Kuramoto et al., 2011). NH<sub>4</sub><sup>+</sup> in surface snow reached maximum values at St. 5 (Fig. S4c). NH<sub>4</sub><sup>+</sup> in the snowpack at St. 3 and St. 9 exhibited minimum values from spring-summer and maximum values from autumn-winter (Fig. 5). Positive peak values of NH<sub>4</sub><sup>+</sup> were exhibited at depths of 1.38, 2.04, and 3.31 m corresponding to autumn-winter periods from 2021-2022, 2020-2021, and 2019-2020, respectively (Fig. 5). According to the stratigraphy of the snowpack at St. 9 (Fig. S2), we attributed the peaks at 1.38, 2.04, and 3.31 m to the deposition of atmospheric ammonium, without the influence of melting and refreezing process. At several sites in the inland of the Greenland Ice Sheet, the concentration of NH<sub>4</sub> peaked from spring-summer owing to biomassburning events (Dibb et al., 2007; Nakazawa et al., 2021; Kjær, et al., 2022). In contrast, the concentrations of NH<sub>4</sub><sup>+</sup> at St. 9 did not peak in spring-summer. The autumn-winter maximum of NH<sub>4</sub><sup>+</sup> at St. 3 and St. 9 have not been observed at the inland sites of the Greenland Ice Sheet. According to an incubation experiment of nitrogen-fixing organisms using artificial seawater, a substantial increase in the atmospheric emission of ammonia gas and gas-phase basic water-soluble organic nitrogen (WSON) was found during the decline and death phases of the organisms (Dobashi, 2023). The NOW polynya facing our study sites is one of the most biologically productive marine areas of the Arctic Ocean (Klein et al., 2002; Odate et al., 2002). Phytoplankton blooms in the NOW onsets in April, are sustained for three months, and then decline and death from autumn-winter (Mei et al., 2002; Marchese et al., 2017). We suggest that the increase in NH<sub>4</sub><sup>+</sup> in the snowpack at St. 3 and St. 9 from autumn–winter was caused by the increase in sea-to-air emission of ammonia gas during the decline and death phases of phytoplankton in the NOW polynya.

Figure 5: Vertical profiles of NH<sub>4</sub><sup>+</sup> at (a) St. 3 and (b) St. 9. Green lines denote mean NH<sub>4</sub><sup>+</sup> across all observation depths. Orange and brown lines denote the mean NH<sub>4</sub><sup>+</sup> plus and minus one standard deviation across all observation depths, respectively. The LOD of NH<sub>4</sub><sup>+</sup> was < 0.0055  $\mu$ eq L<sup>-1</sup>.

## 3.2.4 nssCa<sup>2+</sup>

The possible source of nssCa<sup>2+</sup> is terrestrial dust (Dibb et al., 2007; Kuramoto et al., 2011). In surface snow, the maximum value of nssCa<sup>2+</sup> occurred at St. 2 (Fig. S4f). Positive peak values of nssCa<sup>2+</sup> were exhibited at depths of 0.89, 1.07, and 1.53 m corresponding to early and late spring–summer period in 2021 and 2022 (Fig. 6). According to the stratigraphy of the snowpack at St. 9 (Fig. S2), we attributed the peaks at 0.89, 1.07, and 1.53 m to the deposition of atmospheric nssCa<sup>2+</sup>, without the influence of melting and refreezing process. In the inland areas of the Greenland Ice Sheet, mineral dust, which originates mainly from Asia and North Africa (hereafter referred to as remote dust), was observed in spring layers in snowpacks or ice cores (Steffensen et al., 1988; Whitlow et al., 1992; Mosher et al., 1993; Drab et al., 2002; Kuramoto et al., 2011; Nakazawa et al., 2021; Nagatsuka et al., 2021). The early spring–summer peaks of nssCa<sup>2+</sup> in the snowpack at St. 9 could have originated from remote dust. However, the late spring–summer peaks of nssCa<sup>2+</sup> in the snowpack at St. 9 have not been observed in the inland of the Greenland Ice Sheet. Recently, the decreasing snow cover area in the Arctic associated with Arctic warming, which is processing at a rate two to four times faster than the global average, has increased mineral dust emissions from bare land in the Arctic during the summer–autumn period (hereafter referred to as local dust) (Amino et al., 2020; Matsui et al., 2024). We assumed that the late spring–summer peaks of nssCa<sup>2+</sup> in the snowpack at St. 9 could be attributed to local dust on the northwestern coast of Greenland.

Figure 6: Vertical profiles of  $Ca^{2+}$  and  $nssCa^{2+}$  at (a) St. 3 and (b) St. 9. Green lines denote mean  $nssCa^{2+}$  across all observation depths. Orange and brown lines denote the mean  $nssCa^{2+}$  plus and minus one standard deviation across all observation depths, respectively. The black and red curves denote the total concentration and non-sea-salt (nss) fractions, respectively. The gray lines in (a) and (b) denote values of 0  $\mu$ eq.  $L^{-1}$ . The LOD of  $Ca^{2+}$  was < 0.0050  $\mu$ eq  $L^{-1}$ .

#### $3.2.5 \text{ NO}_3^-$

The possible sources of NO<sub>3</sub><sup>-</sup> are fossil fuel combustion, biogenic soil emission, biomass burning, and photochemical reactions due to lightning (Legrand and Mayewski, 1997; Hastings et al., 2004). The concentration of NO<sub>3</sub><sup>-</sup> in surface snow on Clements Glacier decreased with increasing distance toward the top of the glacier (Fig. S4h). The average concentrations of NO<sub>3</sub><sup>-</sup> at depths from 0.00–1.01 m at St. 3 were greater than that at St. 9 (Table 1). The decreasing trend with distance from the seacoast was similar to that in the eastern region of Prudhoe Land (Matoba et al. 2014). The peak of NO<sub>3</sub><sup>-</sup> at St. 3 was observed on the snow surface corresponding to the spring of 2023 (Fig. 7). The concentrations of NO<sub>3</sub><sup>-</sup> at St. 9 showed several peaks at depths of 0.01, 1.07, 1.94, 2.37, 2.87, 3.15, and 3.57 m (Fig. 7). According to the stratigraphy of the snowpack at St. 9 (Fig. S2), we attributed the peaks at 0.01, 1.07, 1.94 and 3.15 m to the deposition of atmospheric nitrate and the peaks at 2.37, 2.87 and 3.57 m to melting and refreezing process. The NO<sub>3</sub><sup>-</sup> tend to move easily with meltwater and become concentrated during refreezing (Matoba et al., 2002b). Excluding peaks in the ice layers, the seasonal variations in the snowpack at St. 3 and St. 9 exhibited maximum values from late autumn—winter and early spring—summer and low values in

late spring—summer. At several inland sites in Greenland, positive peaks of NO<sub>3</sub><sup>-</sup> occurred in summer and from winter—spring (Kuramoto et al., 2011; Oyabu et al., 2016; Nakazawa et al., 2021). On the basis of the seasonal variation in the nitrogen and oxygen isotopes of NO<sub>3</sub><sup>-</sup> in the snowpack at Summit, the source of NO<sub>3</sub><sup>-</sup> in summer is natural NO<sub>x</sub> produced by biomass burning, biogenic soil emissions, and photochemical reactions due to lightning, whereas that in winter is anthropogenic NO<sub>x</sub> from fossil fuel combustion (Hastings et al., 2004). We assumed that the snowpack at St. 3 and St. 9 included aerosols originating from fossil fuel combustion and did not include aerosols originating from biomass burning, unlike the inland sites in Greenland.

Figure 7: Vertical profiles of  $NO_3^-$  at (a) St. 3 and (b) St. 9. Green lines denote mean  $NO_3^-$  across all observation depths. Orange and brown lines denote the mean  $NO_3^-$  plus and minus one standard deviation across all observation depths, respectively. The LOD of  $NO_3^-$  was  $< 0.0016 \, \mu eq \, L^{-1}$ .

# 3.2.6 nssSO<sub>4</sub><sup>2-</sup>

The possible sources of  $nssSO_4^{2^-}$  are volcanic eruption, fossil fuel combustion, mineral dust, and DMS emissions produced by marine phytoplankton (Jaffrezo et al., 1994; Legrand and Mayewski, 1997). The concentrations of  $nssSO_4^{2^-}$  in surface snow increased from St. 1 to St. 9 (Fig. S4i). The increasing trend with distance from the seacoast was similar to that in the eastern region of Prudhoe Land (Matoba et al. 2014). The concentrations of  $nssSO_4^{2^-}$  at St. 9 showed several positive peaks at depths of 0.01, 1.04, 2.37, 2.68, 2.87, 2.96, and 3.57 m (Fig. 8). According to the stratigraphy of the snowpack at St. 9 (Fig. S2), we attributed the peaks at 0.01, 1.04, 2.68 and 2.96 m to the deposition of atmospheric sulfate and the peaks at 2.37, 2.87 and 3.57

m to melting and refreezing process (Fig. 8). The seasonal variations in the snowpack at St. 3 and St. 9 peaked from late autumn—winter and early spring—summer, excluding positive peaks in the ice layers. At several sites in Greenland, nssSO<sub>4</sub><sup>2-</sup> exhibited maximum values from winter—spring, which is attributed to anthropogenic fossil fuel combustion (Dibb et al., 2007; Kuramoto et al., 2011; Oyabu et al., 2016). Therefore, we believe that the source of positive nssSO<sub>4</sub><sup>2-</sup> peaks in winter and spring at St. 9 could be fossil fuel combustion, such as that at the inland sites in Greenland.

Several negative nssSO<sub>4</sub><sup>2-</sup> peaks were found from autumn–winter in the snowpack at St. 3 and St. 9 (Fig. 8). In a coastal Antarctic ice core, negative peaks have also been observed. The negative values of nssSO<sub>4</sub><sup>2-</sup> in the snowpack result from the low SO<sub>4</sub><sup>2-</sup>/Na<sup>+</sup> in aerosols emitted from frost flowers on newly formed sea ice (Rankin et al., 2002; Woolf et al., 2003; Rankin et al., 2004). Frost flowers are depleted in sulfate relative to sodium owing to the precipitation of mirabilite (Na<sub>2</sub>SO<sub>4</sub>10H<sub>2</sub>O) during the formation of sea ice at temperatures below –8 °C (Richardson, 1976; Rankin et al., 2000; Rankin et al., 2002; Hara et al., 2017). In northwestern Greenland, the frost flowers collected from newly formed sea ice in front of Siorapaluk indicated depletion in sulfate relative to that in seawater (Hara et al., 2017). The negative nssSO<sub>4</sub><sup>2-</sup> peaks observed at our study sites could be attributed to the atmospheric emission of frost flowers from newly formed sea ice in the NOW polynya, where new sea ice is often created and broken up by strong northerly winds in winter.

Figure 8: Vertical profiles of  $SO_4^{2-}$  and  $nssSO_4^{2-}$  at (a) St. 3 and (b) St. 9. Green lines denote mean  $nssSO_4^{2-}$  across all observation depths. Orange and brown lines denote the mean  $nssSO_4^{2-}$  plus and minus one standard deviation across all observation depths, respectively. The black and red curves denote the total concentration and non-sea-salt (nss) fractions, respectively. The gray lines in (a) and (b) denote values of 0  $\mu$ eq. L<sup>-1</sup>. The LOD of  $SO_4^{2-}$  was < 0.0021  $\mu$ eq L<sup>-1</sup>.

#### 3.2.7 MSA

MSA is an oxidation product of DMS, which originates from ocean phytoplankton (Charlson et al., 1987; Jaffrezo et al., 1994). The concentration of MSA in surface snow reached a maximum value at St. 6 (Fig. S4j). At St. 3, an exceptionally high MSA was observed at a depth of 1.01 m, corresponding to the summer of 2022 (Fig. 9a). The snowpack at St. 9 exhibited multiple distinct MSA peaks at depths of 0.83, 1.07, 2.68, 2.91, and 3.57 m (Fig. 9b). According to the stratigraphy of the snowpack at St. 9 (Fig. S2), we attributed the peaks at 0.83, 1.07, and 2.68 m to the deposition of atmospheric MSA and the peaks at 2.91 and 3.57 m to melting and refreezing processes. The positive MSA peak at 1.07 m coincided with the depth of the positive NO<sub>3</sub><sup>-</sup> and nssSO<sub>4</sub><sup>2</sup><sup>-</sup> peaks. This finding indicates that MSA originating from ocean phytoplankton activity at middle and low latitudes could have been transported to St. 9 with anthropogenic products, including NO<sub>3</sub><sup>-</sup> and nssSO<sub>4</sub><sup>2</sup>. In the snowpack at depths of 0.83 and 2.63 m corresponding to spring—summer layers, MSA and/or nssSO<sub>4</sub><sup>2</sup> showed positive peaks, whereas NO<sub>3</sub><sup>-</sup> did not indicate any peaks. Because NO<sub>3</sub><sup>-</sup> peaks did not occur, the air masses containing MSA were not transported from mid-latitudes, which are areas of human activity. Therefore, the positive peaks of MSA at 0.83 and 2.63 m could be reflected in sea-to-air DMS emission attributed to the ocean phytoplankton bloom in the NOW polynya.

Figure 9: Vertical profiles of MSA at (a) St. 3 and (b) St. 9. Green lines denote mean MSA across all observation depths. Orange and brown lines denote the mean MSA plus and minus one standard deviation across all observation depths, respectively. The LOD of MSA was  $< 0.0011 \, \mu eq \, L^{-1}$ .

422

425

429

436437

441442

444445

#### 3.3 Intense poleward heat and moisture transport events in winter from 2022–2023

The concentrations of  $\delta^{18}$ O, Na<sup>+</sup>, Cl<sup>-</sup>, K<sup>+</sup>, Mg<sup>2+</sup>, SO<sub>4</sub><sup>2-</sup>, and Ca<sup>2+</sup> increased substantially and the nssSO<sub>4</sub><sup>2-</sup> exhibited negative values at depths from 0.26-0.42 m at St. 3 and St. 9, corresponding to autumn-winter from 2022-2023 (Fig. 10), According to the stratigraphy of the snowpack at St. 3 and St. 9, the peaks of  $\delta^{18}$ O and ion concentrations observed from 0.26–0.42 m had not been changed by the melting and refreezing process. Because the depth of the positive peaks of  $\delta^{18}$ O and these ions coincided between St. 3 and St. 9, we believe that the snowpacks from 0.26–0.42 m originated from the same precipitation event during the autumn-winter period from 2022-2023. In previous observations of precipitation isotopes in Siorapaluk, the  $\delta^{18}$ O value in precipitation was significantly correlated with the surface air temperature from winter-spring (Matoba et al., 2002a). Therefore, the rapid positive increase in  $\delta^{18}$ O at depths from 0.26–0.42 m at St. 3 and St. 9 suggested that the air temperature at our study sites increased during low temperature period, leading to the warmest in winter from 2022-2023. Continuous weather monitoring in Siorapaluk revealed that the air temperature increased by more than 0 °C and that the southern component of the wind increased from 5-6 December 2023 (Fig. 11b and c). During this period, the atmospheric pressure gradient at 500 hPa increased along Baffin Bay due to high pressure in Greenland and low pressure in northern Canada. and the southerly wind speed along the western coast of Greenland increased during this period (Fig. 11d and Fig. 12). Consequently, we determined that the rapid increase in  $\delta^{18}$ O at depths from 0.26–0.42 m corresponded to the intense poleward heat and moisture transport event from 5-6 December 2023. The Na<sup>+</sup>, Cl<sup>-</sup>, K<sup>+</sup>, Mg<sup>2+</sup>, SO<sub>4</sub><sup>2-</sup>, and Ca<sup>2+</sup> concentrations increased substantially, and nssSO<sub>4</sub><sup>2-</sup> exhibited negative values at depths from 0.26–0.42 m at St. 3 and St. 9 (Fig. 10). The Na<sup>+</sup>, Cl<sup>-</sup>, K<sup>+</sup>,  $Mg^{2+}$ ,  $SO_4^{2-}$ , and  $Ca^{2+}$  concentrations in frost flowers are comparable to or greater than those in seawater, and the  $SO_4^{2-}/Na^+$ ratio of frost flowers is lower than that of seawater (Hara et al., 2017). On the eastern side of the NOW, the southern component of the wind at 10 m a.g.l. increased from 5-6 December 2023, whereas the sea ice concentration remained high (Fig. 11a and c). The increase in the surface wind speed across the sea ice in the NOW from 5-6 December 2023 promoted the atmospheric emission of frost flowers from sea ice, after which the Na<sup>+</sup>, Cl<sup>-</sup>, K<sup>+</sup>, Mg<sup>2+</sup>, SO<sub>4</sub><sup>2-</sup>, and Ca<sup>2+</sup> concentrations increased, and nssSO<sub>4</sub><sup>2-</sup> exhibited negative values at depths from 0.26–0.42 m at St. 3 and St. 9. We suggest that intense poleward heat and moisture transport events attributed to the high pressure in Greenland and low pressure in northern Canada caused precipitation, including heavier isotopes and frost flowers on the northwestern coast of Greenland Ice Sheet.

Figure 10: Vertical profiles of  $\delta^{18}$ O, Na<sup>+</sup>, Cl<sup>-</sup>, K<sup>+</sup>, Mg<sup>2+</sup>, Ca<sup>2+</sup>, SO<sub>4</sub><sup>2-</sup>, and nssSO<sub>4</sub><sup>2-</sup> at (a–h) St. 3 and (i–p) St. 9. Green lines denote mean ion concentrations across all observation depths. Orange and brown lines denote the mean ion concentrations plus and minus one standard deviation across all observation depths, respectively.

Figure 11: Sea ice and meteorological conditions from June 2022 to May 2023 corresponding to snow depths ranging from 0.00–1.15 m at St. 9. (a) shows the sea ice concentration on the eastern side of NOW. (b) shows the temperature at 2 m a.g.l. in Siorapaluk (red line) and St. 9 (black line). (c) shows the southern component of the wind speed at 2 m a.g.l. in Siorapaluk (red line) and 10 m a.g.l. on the eastern side of the NOW (black line). (d) shows the meridional wind speed at 700 hPa along the eastern side of Baffin Bay. The values on the eastern side of the NOW were averaged over 75.0°–80.0°N and 75.0°–65.0°W. The values on the eastern side of Baffin Bay were averaged over 75.0°–65.0°W from 75.0° to 80.0°N, 75.0°–65.0°W from 75.0° to 80.0°N, 75.0°–65.0°W from 75.0° to 80.0°N. The positive and negative values of the meridional wind speed are the southern and northern parts of the wind, respectively. The black arrows denote the date from 5–6 December 2023.

Figure 12: Geopotential height at 500 hPa averaged from 5 to 6 December 2022. The white circle indicates the location of St.9.

## **4 Conclusion**

We conducted glaciological observations from 9–11 April 2023 on the western side of Prudhoe Land in northwestern Greenland facing the NOW to elucidate the source conditions and transportation processes of water vapor and aerosols in this region. The dating of the snowpack at St. 9, which is located at the inland of the western side of Prudhoe Land, revealed that the layer at a depth of 4.20 m corresponded to 3.5 years. The average annual accumulation at St. 9 was 0.49 m w.eq. yr<sup>-1</sup>.

The snowpacks on the western side of Prudhoe Land contained aerosols from distant sources, such as remote dust and anthropogenic aerosols, in early spring—summer layers. On the other hand, they also contained aerosols from local sources such as ocean biological activity and frost flowers in the NOW and local dust around the coast of northwestern Greenland during other seasons, unlike the inland of the Greenland Ice Sheet. Moreover, we noted that the snowpacks were able to trace the poleward heat and moisture transport event along Baffin Bay during winter.

Arctic climate warming caused decreases in the sea ice thickness and concentration over the last few decades in the NOW and could influence clouds and precipitation following changes in sea ice and biological activities in the NOW. We found for the first time that the environmental changes in the NOW can be elucidated by the snowpack and ice core on the western side of the Prudhoe Land. We suggest that the chemical substances in the deeper ice core from this region could help explain the multidecadal variations in the sea ice, biological activities, and related water and aerosol circulation around the NOW and could develop to understand the accurate future projections of environmental change in this region.

## 483 Code availability

- Data analyzes were performed using Interactive Data Language (IDL) (version 8.7.1) developed by Exelis Visual
- Information Solutions, Boulder, Colorado. Map figures were generated using Generic-Mapping Tools (GMT) (version 6.1.0)
- (https://www.generic-mapping-tools.org/) and Quantum Geographic Information System (QGIS) (https://qgis.org/).

## 487 Data availability

- The data used in this are available from the Hokkaido University Collection of Scholarly and Academic Papers (HUSCAP)
- (http://hdl.handle.net/2115/94742). The AWS data are available from the Arctic Data archive System (ADS)
- (https://ads.nipr.ac.jp/data/meta/A20241031-013). The ERA5 and ERA5-Land data are available from the Copernicus Climate
- Change Service (C3S) Climate Data Store (https://cds.climate.copernicus.eu/#!/search?text=ERA5&type=dataset). The
- Landsat-8 image is provided by United States Geological Survey (https://earthexplorer.usgs.gov/). The Greenland Mapping
- Project 2 (GIMP-2) digital elevation model version 2 is provided by the National Snow and Ice Data Center (NSIDC)
- (https://nsidc.org/data/nsidc-0715/versions/2).

#### 495 Author contributions

- Y.K. and S.M. designed this study. Y.K., M.M, T.K, and H.I. collected and processed the snow samples during the field
- observations. Y.K. and S.M. analyzed water stable isotopes and chemical substances in the snow samples. Y.K. analyzed the
- meteorological and sea ice data. Y.K. and S.M. wrote the manuscript draft. Y.I. revise the manuscript.

#### 499 Competing interests

The contact author has declared that none of the authors has any competing interests.

#### 501 Acknowledgement

- We thank Dr. Akihiro Hashimoto of Meteorological Research Institute, Japan Meteorological Agency, for the provided
- ground support in Japan. We also thank Yusuke Kakuhata and Ikuo Oshima for the valuable suggestions for the field
- observations.

## 505 Financial support

- This study was supported in part by (1) the Japan Society for the Promotion of Science (JSPS) KAKENHI grant 22J10351,
- 18H05292, 23H00511, and 24H00757, (2) the Arctic Challenge for Sustainability (ArCS II) Project (JPMXD1420318865),
- and (3) the National Institute of Polar Research through Special Collaboration Project no. B24-02.

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
