# Peer review of "Characteristics of snowpack chemistry on the coastal region in the"

_EGUsphere, 2025_

## Referee Comment (RC2)

**Review Comments**

**General Comments**

This manuscript presents glaciological observations conducted from the coast to the inland areas of Prudhoe Land, located in the northwestern Greenland Ice Sheet. The primary objective of the study is to enhance understanding of water and aerosol circulation in the North Water (NOW) region. Using surface snow and snow pit samples, the study investigates variations in water isotopic composition and ionic species concentrations. This research is a valuable preliminary step toward reconstructing past environmental changes in the NOW region using ice cores. A better understanding of historical environmental changes in this area is expected to improve future projections of marine environmental shifts in the region.

However, several aspects of the study's results require further consideration.

First, data from the ST9 snow pit indicate evidence of summer surface snowmelt. Such melting processes hinder the preservation of proxy records and introduce uncertainty in age-dating. As discussed in the manuscript (lines 164–165), water isotope records tend to become smoothed, and ion concentrations are altered due to refreezing of meltwater. Therefore, the interpretation of vertical variations in proxy concentrations should account for these site-specific characteristics.

Particularly in Section 3.2 ("Spatial and temporal variations in water isotopes and chemical species"), the interpretation of ST9 data should reflect the impact of summer melt on concentration variability.

The seasonal classification such as spring–summer vs. autumn–winter should be used consistently, and the discussion of concentration variability should be supported by statistical criteria due to no clear variability of proxies. For example, it is recommended to define peaks using either values above the mean or above the mean plus one standard deviation.

Second, additional evidence is required to substantiate some of the manuscript's interpretations. For example, to support the discussion on atmospheric transport, the inclusion of backward trajectory modeling (e.g., frequency maps and cluster analyses) is recommended as supplementary information to identify source regions and air mass pathways.

**Specific Comments**

**Introduction**

- Lines 44–70: The necessity of studying past environmental changes in the NOW region is well presented. However, further explanation is needed on how the current study site differs from the nearby SIGMA-A site, especially in terms of meteorological conditions like prevailing wind directions.
- Line 68–69: Rephrase for clarity.

**Methods**

- Lines 104–105: Add information in the Supplementary Information regarding the design and cleaning procedures of the pre-cleaned stainless-steel tools used for snow pit sampling. Clarify the cleanliness specification of the Whirl-Pak polyethylene bags (e.g., part number, manufacturer).
- Because sample depth resolution varies (2 cm, 3 cm, 5–10 cm), figures such as Figure 4 should adopt a step-wise format for clarity, not dot and line format.
- Line 112: Provide details about possible contamination during snow sample melting and bottling. if possible, field blank should be provided.
- Line 117: Include specifications of the analytical column (e.g., length, diameter), model/manufacturer of standard materials, and detection limits for each ion.
- Line 122: Specify the standard material used for stable water isotope analysis.

**Results and Discussion**

- Line 139: Present snow density alongside depth.
- Lines 143–144: Ice layers below 0.96 m in the ST9 snowpack suggest summer melting, which may affect proxy preservation. This is appropriately and kindly described in lines 163–168.
- Line 172: Calculate annual accumulation rates using snow density for each depth interval and present average values.
- Line 183: Indicate the MSA detection limit as a line in Figure 4. Clarify dating below 3.4 m at ST9 (the conclusion mentions dating down to 4.5 m).
- Line 188: Include $NO_3^-$ data.
- Line 196: Use nss-$Ca^{2+}$ to interpret dust transport. Since nss-$K^+$ and nss-$Mg^{2+}$ mostly show negative values, suggesting major marine influence, omit these from discussion and Table 1.
- Lines 197–209: Explain shortly the notable difference in $\delta^{18}O$ between the upper layer (0–0.7 m) and the deeper layer.
- Line 201: Present backward trajectory modeling results to support atmospheric transport path interpretations.
- Line 216: Interpretation in Figure 6c should align with the seasonal framework in Figure 6b.
- Lines 222, 232: Revise for clarity.
- Line 239: Provide supporting data for air mass transport.

- Line 278: Explain nitrate concentration increases due to melting/refreezing. if possible, explain shortly or provide references. Revise "positive peaks" to just "peaks."
- Table 1: Replace nss-$K^+$ with $K^+$ and nss-$Mg^{2+}$ with $Mg^{2+}$ data.
-

**Conclusion**
- Avoid repeating earlier content. Summarize only the most significant findings and implications.

---

## Author Comment (AC1)

**Dear Reviewer, #1**

**Thank you very much for your valuable comments. Our responses and the changes that we plan to make in the revised manuscript are explained below. We filled in reviewer comments in black, author replies in blue, the proposed changes to the revised manuscript in red.**

**Reviewer comment:**

In this paper entitled "Characteristics of snowpack chemistry on the coastal region in the northwestern Greenland Ice Sheet facing the North Water", authors present an interesting observation of the effect that polynia North water (NOW) has on aerosol circulation and precipitation. the results are obtained from measurements of major ions, MSA and water isotopic analyses at 9 surface snow sampling sites, 2 snow-pit sites and 1 ice core. The text is well structured a detailed introduction, however the drafting in general should be improved as there are numerous repetitions and in some parts the reading is difficult to understand. In particular, the section 3.2 has to be improved. The conclusions have to be focused on the main goals obtained in this paper. It is very long and I suggest to summarize, avoiding to repeat the results and discussion.

**Author reply:**

To improve the overall logical flow and readability in section 3.2, we will add individual sub-sections for $\delta^{18}O$ and ion species. Figures of $\delta^{18}O$ and each ion concentration will be presented separately within their respective sub-sections.

We will summarize the conclusion section as follows and avoid some repetitions.

=====

**4 Conclusion**

We conducted glaciological observations from 9−11 April 2023 on the western side of Prudhoe Land in northwestern Greenland facing the NOW to elucidate the source conditions and transportation processes of water vapor and aerosols in this region. The dating of the snowpack at St. 9, which is located at the inland of the western side of Prudhoe Land, revealed that the layer at a depth of 4.20 m corresponded to 3.5 years. The average annual accumulation at St. 9 was 0.49 m w.eq. yr$^{-1}$.

The snowpacks on the western side of Prudhoe Land contained aerosols from distant sources, such as remote dust and anthropogenic aerosols, in early spring‑summer layers. On the other hand, they also contained aerosols from local sources such as ocean biological activity and frost flowers in the NOW and local dust around the coast of northwestern Greenland during other seasons, unlike the inland of the Greenland Ice Sheet. Moreover, we noted that the snowpacks were able to trace the poleward heat and moisture transport event along Baffin Bay during winter.

Arctic climate warming caused decreases in the sea ice thickness and concentration over the last few decades in the NOW and could influence clouds and precipitation following changes in sea ice and biological activities in the NOW. We found for the first time that the environmental changes in the NOW can be elucidated by the snowpack and ice core on the western side of the Prudhoe Land. We suggest that the chemical substances in the deeper ice core from this region could help explain the multidecadal variations in the sea ice, biological activities, and related water and aerosol circulation around the NOW and could develop to understand the accurate future projections of
environmental change in this region.
=====

**Reviewer comment:**

On lines105-106: "The snow sampling intervals at St. 3 were 0.02 m from 0.00 to 0.20 m and 0.03 m from 0.20 to
1.01 m, and the snow sampling intervals at St. 9 were 0.02 m from 0.00 to 0.20 m and 0.03 m from 0.30 to 1.08
m." Why was the sampling interval changed?

**Author reply:**

If we could sample the entire snowpack at short intervals, we would have been able to discuss the temporal
variations in chemical components with short time intervals. However, we changed sampling interval partway
through the snowpack, because we had limitation on the number of snow samples that could be transported by dog
sledges.

**Reviewer comment:**

Lines 104, 108, 110. The authors told of precleaned materials and tools, but the cleaning procedure is not described.

**Author reply:**

We will add the cleaning procedure to the method section as follows.
=====
We removed the oil contamination on the precleaned materials and tools using ethanol, and performed then
ultrasonic cleaning in ultrapure water.
=====

**Reviewer comment:**

On line 107: Why was the ice core only sampled at one site? could be used for comparison at least with st3.

**Author reply:**

We prioritized sampling as much as possible at St. 9 because of the limitation on the number of snow samples that
could be transported by dog sledges.

**Reviewer comment:**

Line 115: "methane sulfonate− (hereafter referred to as MSA)" already defined in the introduction

**Author reply:**

The definition of MSA will be moved to the Introduction.

**Reviewer comment:**

Lines 116-118. Please add several details about the analytical methods or some references. In particular, the authors declared only the columns used for cations and anions without any specific important details such as dimensions. Other important details are flows, injection volumes, instruments used, suppressors, detectors. No specific details about the quantification methods are reported. I suppose that you used external calibration curves, but which are the linear ranges, and which are the RCM used for quantification. In summary, please improve the method and quality control section about the ionic analysis.

**Author reply:**

We will add details about the analytical method of the ion chromatography in the method section as follows.

=====

For the cations, separation was carried out with a Dionex CG-12 (4 × 50 mm) guard column, followed by a Dionex CS12-A (4 × 250 mm) separation column. Injection volume of samples was 500 µL. MSA (20 mM) was used as eluent, and flow-rate was kept 1.0 mL min-1. Dionex CDRS600 dynamically regenerated suppressor was used for conductivity suppression before conductivity cell. For the anions, separation was obtained with a Dionex AG-18 (4 × 50 mm) guard column and Dionex AS-18 (4 × 250 mm) separation column. Injection volume of samples was 1000 µL. KOH (23 mM) was used as eluent, and flow-rate was kept 1.0 mL min-1. Dionex ADRS 600 dynamically regenerated suppressor was used for conductivity suppression before conductivity cell. 5-point calibration curves were used for quantitative determination of each ion. The calibration curves were constructed using standard solution (Fujifilm Waco) adjusted to 20, 50, 100, and 200 ppb with ultra-pure water. If the ion concentration of samples were outside the calibration range (> 200 ppb), it was remeasured using 500, 1000, 2000~3000, and 6000 ppb standard for the anions and 500, 1000, 2000, and 4000 ppb standard for the cations. Blanks were always evaluated before the calibration procedure.

=====

**Reviewer comment:**
Lines 117-119: Has the ion chromatography method used been validated in previous works? If yes, indicate them, if not, insert a section on validation.

**Author reply:**
The ion chromatography method had been validated by the previous work (Kurosaki et al., 2020; Kurosaki et al., 2022). We will add this description to the method section as follows.

=====

The ion chromatography method in this study had been evaluated by our previous study (Kurosaki et al., 2020; Kurosaki et al., 2022).

=====

**Reviewer comment:**

Lines 119-120: "The samples exhibiting large peak were measured multiple times, to confirm that any large peak in ion concentration was not caused by analytical errors." What is meant?

**Author reply:**

The large peaks of $Na^+$ and $Cl^-$ were outside the calibration range up to 200 ppb. These samples were remeasured using 500, 1000, 2000~3000, and 6000 ppb standard for the anions and 500, 1000, 2000, and 4000 ppb standard for the cations.

We will add this text to the method section as follows.

=====

The 5-point calibration curves were constructed using standard solution (Fujifilm Waco) adjusted to 20, 50, 100, and 200 ppb with ultra-pure water. If the ion concentration of samples were outside the calibration range (> 200

ppb), it was remeasured using 500, 1000, 2000~3000, and 6000 ppb standard for the anions and 500, 1000, 2000, and 4000 ppb standard for the cations. Blanks were always evaluated before the calibration procedure.

=====

**Reviewer comment:**

Lines 156-165 Text is not clear

**Author reply:**

We have revised the text you kindly pointed out as follows.

=====

We determined the seasonality of the snowpack from 0.00 to 1.15 m as shown below. The date of snow layer from

0.00–0.04 m was close to the observation date (April 2023). We determined that snowpack in 0.04–0.72 m corresponded to the autumn to winter period from 2022–2023 from the negative $\delta^{18}O$ peak, positive d-excess peak and low MSA values (Fig. 2). The snowpack from 0.72 m–1.15 m corresponded to spring to summer in 2022 from existence of ice layer, high $\delta^{18}O$ value and high MSA values (Fig. 2 and Fig. S2). Snowpack below 0.96 m corresponded to previous summer and before it because the amplitude of seasonal variation of $\delta^{18}O$ and d-excess below 0.96 m were smaller than those in the shallower layers from 0.00–0.96 m because of summer melting (Fig.

2a and b). The MSA concentration showed obvious seasonal variations, and the $\delta^{18}O$ values exhibited slight seasonal variations below 0.96 m (Fig. 2a and c), although the values of the water stable isotopes were smoothed by summer melting and some chemical species showed high peaks in the ice layers owing to relocation processes by meltwater refrozen.

=====

**Reviewer comment:**

Section 3.2. Following stratigraphic analysis and evaluation of snowpack density, it may be more informative to express data in terms of fluxes rather than concentrations, so in the subsequent data analysis one could avoid distinguishing peaks attributed to atmospheric deposition from those of melting and refreezing

**Author reply:**

As you have pointed out, the deposition flux is sometimes more suitable when discussing the deposition amount of atmospheric aerosols for quantitatively. However, we cannot discuss the deposition flux because we did not collect the snow density with high resolution along the snow depth. Therefore, we qualitatively discussed the seasonal characteristics of ion species based on their concentration.

**Reviewer comment:**

Lines 188–190: Introducing all figures at the beginning of the section may lead to confusion. Since the discussion begins with Fig. 5, it would be more effective to present the figures sequentially, in alignment with the narrative.

**Author reply:**

In accordance with your comment, we will revise the order of figures. We will present the $\delta^{18}O$ and each ion concentration within the relevant sub-sections of section 3.2, displaying the figures sequentially.

**Reviewer comment:**

Line 194: "We applied the concentration unit as µeq L−" Information that is already made explicit in the following graphs

**Author reply:**

This sentence notes that "µ eq L$^{-1}$" was used as the unit of ion concentration in equation (1). The editor requested that the concentration unit should state clearly for the equation (1).

**Reviewer comment:**

Line 201: "We suggest that the spatial variation in the $\delta^{18}O$ results from water vapor transport from the southern coast to the northern inland area by southerly winds." Might it be useful to indicate figure 9 by referring to the direction of the prevailing winds?

**Author reply:**

Thank you for your comment. We suggested that the south-to-north gradient of the $\delta^{18}O$ results from water vapor from the southern coast to northern inland area by the southerly winds. We have performed the backward trajectory analysis and analyzed the probability map of air mass transportation to make this assumption more reliable (Fig. 1

in this file). The 7-days backward trajectory of air mass arriving at St. 9 showed that the majority of air mass was transported from the south of St. 9, situated on northern Baffin Bay and eastern NOW.

We will add the method of backward trajectory in section 2.3 as follows.

=====

**2.3 Backward trajectory**

To investigate the source region and transport pathway of water vapor and aerosols contained in ice core at the St. 9 site, we analysed air mass position along the backward trajectory from the St. 9 site during the past 7 days using the National Oceanographic and Atmospheric Administration (NOAA) Hybrid Single-Particle Lagrangian Integrated Trajectory (HYSPLIT) model (Stein et al., 2015) and National Centers for Environmental Prediction (NCEP) reanalysis data. The initial positions of air mass were set at 50, 500, 1000, 1500 m above ground level over the St. 9 site. The initial date and time were every 6 h from 2019 to 2023. We calculated the probability of the existence of an air mass with a 1° resolution. Considering the water vapor and aerosols supply from the ocean and land surface, we excluded air mass over 1000 m above ground level. The existence probability was weighted by the daily amount of precipitation when the air mass arrived at the St. 9 site. The daily amount of precipitation was extracted from the ERA5 reanalysis dataset (Hersbach et al., 2020).

=====

We will revise the early part of result and discussion of the $\delta^{18}$O (section 3.2.1) as follows. Figure 1 in this file will be added to the supplementary material.

=====

**3.2.1 $\delta^{18}$O**

The spatial variations in $\delta^{18}$O in the surface snow showed maximum and minimum values at St. 3 (−19.12 ‰) and St. 9 (−37.21 ‰), respectively (Fig. S4a). The average $\delta^{18}$O value from 0.00 to 1.01 m at St. 3 was greater than that at St. 9 (St. 3: −22.03 ‰; St. 9: −29.12 ‰) (Table 1). The $\delta^{18}$O values in surface snow and the snowpack decreased from the seacoast toward the inland site. The past 7 days backward trajectory arriving at the St. 9 also exhibited that majority of air mass was transported from the south of the St. 9, situated on the northern Baffin Bay and eastern NOW (Fig. S5). We suggest that the south-to-north gradient of $\delta^{18}$O results from water vapor, which originates from northern Baffin Bay and eastern NOW, transport from the southern coast to the northern inland area by southerly winds.

=====

[Figure]

**Figure 1 (in this file): Existence probability of an air mass occurring during the past during 7 days reaching the St. 9 in whole of year from 2019–2023.** (a) and (b) display Arctic area (> 60°N) and around northwestern Greenland, respectively. Black circles show the position of the St. 9.

**Reviewer comment:**

Line 210. Please add "(figure 6)" to help readers or start the sentence introducing the Figure 6 and its meaning.

**Author reply:**

The explanation of related figure will be added at the beginning of the paragraph about the vertical profile of $\delta^{18}O$ at St. 3 and St. 9 as follows.

=====

Fig. 3 shows the vertical profile of δ¹⁸O in the snowpack at St. 3 and St. 9, and the difference in δ¹⁸O between St.

9 and St. 3. The vertical profile of δ¹⁸O in the snowpacks at St. 9 was similar to that at St. 3.

=====

**Reviewer comment:**

Figure 6: I suggest using the season and year instead of Roman numerals, as this would facilitate interpretation.

This recommendation may also apply to the other figures. It is somewhat difficult to follow the discussion, as it requires frequently switching between different figures.

**Author reply:**

To improve the overall logical flow and readability in section 3.2, we will add individual sub-sections for δ¹⁸O and ion species. Figures of δ¹⁸O and each ion concentration will be presented separately within their respective sub- sections. The seasonal divisions in each figure will be revised from Roman numerals to explicit labels indicating the season and year (ex. Fig. 2 in this file).

[Figure]

**Figure 2 (in this file): Vertical profile of NH₄⁺ at (a) St. 3 and (b) St. 9.** Green lines denote mean NH₄⁺ across all observation depths. Orange and brown lines denote the mean NH₄⁺ plus and minus one standard deviation across all observation depths, respectively. The LOD of NH₄⁺ was < 0.0055 µeq L⁻¹.

**Reviewer comment:**

Figure 6c, it is not clear why the authors used the difference between St3 and St.9, instead of a ratio.

**Author reply:**

To discuss the seasonal variation of the surface air temperature difference between St. 3 and St. 9, we calculated the difference of δ¹⁸O values at the two stations.

We propose that the difference between St. 3 and St. 9 can be discussed for the following reasons. The depths of the negative and positive peaks of δ¹⁸O at St. 9 agreed well with those at St. 3 (Fig. 3 in this file), and the vertical profile of δ¹⁸O between 0.00 and 1.01 m at St. 9 correlated significantly with that at St. 3 ($r = 0.69$, $p < 0.01$). The snowpack corresponding to autumn−winter from 2022−2023 at St. 3 and St. 9 at the same snow depth were most likely accumulated with precipitation attributed to the same snowfall events, and δ¹⁸O in the snowpack had not been changed by the post depositional processes, which is water molecule diffusion, wind blowing, and sublimation. Therefore, we propose that the vertical profile of δ¹⁸O between 0.00 and 1.01 m at St. 9 can be reasonably compared with the profile at St. 3 based on their differences. We have already described the above discussion in our manuscript.

[Figure]

**Figure 3 (in this file): Vertical profile of δ¹⁸O. (a) and (b) show δ¹⁸O values at St. 3 and St.9, respectively.** (c) shows difference between St.3 and St.9 in terms of δ¹⁸O. i–vii denote seasons from 2019 to 2023. i, iii, v, and vii denote from autumn to winter period from 2022–2023, 2021–2022, 2020–2021, and 2019–2020, respectively. ii, iv, and vi denote from spring to summer in 2022, 2021, and 2020, respectively.

**Reviewer comment:**
Line 217-218: "We suggest that the altitude gradient of the surface air temperature in winter was greater than that in summer in the western region of Prudhoe Land." could this statement also be confirmed using atmospheric models for specific sites?

**Author reply:**

We estimated the difference in surface air temperature between St. 9 and St. 3 using ERA5-Land reanalysis dataset (Fig. 4 and Fig. 5 in this file). The temperature difference between St. 9 and St. 3 was greatest in summer. The mean temperature differences in autumn and winter were more negative than that in summer. This result supports our suggestion, based on water stable isotope, that the altitude gradient of surface air temperature in the western side of Prudhoe Land was steeper in winter than in summer.

We will add this description in section 3.2.1 as follows. Figure 4 and 5 in this file will be added to the supplementary material.

 =====

We estimated the difference in surface air temperature between St. 9 and St. 3 using ERA5-Land reanalysis dataset (Fig. S5 and Fig. S6). The temperature difference between St. 9 and St. 3 was greatest in summer. The mean temperature differences in autumn and winter were more negative than that in summer. This result supports our suggestion, based on water stable isotope, that the altitude gradient of surface air temperature in the western side of Prudhoe Land was steeper in winter than in summer.

=====

We will add the explanation of ERA5-Land in the method section as follows.

=====

Additionally, we used 2 m air temperature in the western side of the Prudhoe Land from ERA5-Land reanalysis dataset supplied by the ECMWF (Muñoz-Sabater et al., 2021)

=====

[Figure]

[Figure]

**Figure 4 (in this file): Diurnal variations in (a) 2 m air temperature at St. 3 and St. 9, and (b) 2 m air**

**temperature difference between St. 9 and St. 3.**

[Figure]

Temperature gradient between St. 9 and St. 3

**Figure 5 (in this file): Seasonal variations in the difference of 2 m air temperature between St. 9 and St. 3.**

**Reviewer comment:**

Lines 306-309: there are many repetitions of "the concentration of MSA". Same in the conclusions with "The snowpack on the western side of Prudhoe Land".

**Author reply:**

Thank you for your kind comment. We will revise the text that you have pointed out as follows.

=====

**3.2.7 MSA**

MSA is an oxidation product of DMS, which originates from ocean phytoplankton (Charlson et al., 1987; Jaffrezo et al., 1994). The concentration of MSA in surface snow reached a maximum value at St. 6 (Fig. S4j). At St. 3, an exceptionally high MSA was observed at a depth of 1.01 m, corresponding to the summer of 2022 (Fig. 9a). The snowpack at St. 9 exhibited multiple distinct MSA peaks at depths of 0.83, 1.07, 2.68, 2.91, and 3.57 m (Fig. 9b).

**4 Conclusion**

We conducted glaciological observations from 9−11 April 2023 on the western side of Prudhoe Land in northwestern Greenland facing the NOW to elucidate the source conditions and transportation processes of water vapor and aerosols in this region. The dating of the snowpack at St. 9, which is located at the inland of the western side of Prudhoe Land, revealed that the layer at a depth of 4.20 m corresponded to 3.5 years. The average annual accumulation at St. 9 was 0.49 m w.eq. yr$^{-1}$.

The snowpacks on the western side of Prudhoe Land contained aerosols from distant sources, such as remote dust and anthropogenic aerosols, in early spring–summer layers. On the other hand, they also contained aerosols from local sources such as ocean biological activity and frost flowers in the NOW and local dust around the coast of northwestern Greenland during other seasons, unlike the inland of the Greenland Ice Sheet. Moreover, we noted that the snowpacks were able to trace the poleward heat and moisture transport event along Baffin Bay during winter.

Arctic climate warming caused decreases in the sea ice thickness and concentration over the last few decades in the NOW and could influence clouds and precipitation following changes in sea ice and biological activities in the NOW. We found for the first time that the environmental changes in the NOW can be elucidated by the snowpack and ice core on the western side of the Prudhoe Land. We suggest that the chemical substances in the deeper ice core from this region could help explain the multidecadal variations in the sea ice, biological activities, and related water and aerosol circulation around the NOW and could develop to understand the accurate future projections of environmental change in this region.

=====

**Reviewer comment:**

General comment on the conclusions: from figure 1 sampling sites 1 to 5 (or 6) are in a valley. has this aspect been taken into consideration? could it have an impact on the final considerations?

**Author reply:**

I appreciate your valuable comment.

Because the topography in the western side of Prudhoe Land is smooth (Fig. 1 in this file) and the glacier is broad and relatively low gradient, we think that the enhancement of vertical convection and downslope wind caused by the valley topography are insignificant on the large-scale water vapor and aerosol circulation around the western side of the Prudhoe Land.

**Other comments:**

**Reviewer comment:**

In figure 1b it might be useful to include a dimensional scale to give an idea of the distances.

Similarly, in figure 2, in addition to the distance expressed in latitude, could a conversion to km be useful?

**Author reply:**

Thank you for your ideas. We have added the scale of distance and north arrow (Fig. 6 in this file), and the distance from St. 1 to each sampling station (Fig. 7 in this file).

[Figure]

**Figure 6 (in this file): Maps of the sampling sites.** (a) shows location of the snowpit and ice core sampling sites in this study (St. 9) and previous studies (SIGMA-A, SIGMA-D, and NEEM) in the northwestern Greenland Ice Sheet. The dashed polygon in (a) denotes the approximate location of the NOW. Hayes peninsula in the northwestern Greenland is located between Kane Basin in the north and Melville Bay in the south. (b) shows Landsat-8 image around St. 9 and SIGMA-A of Prudhoe Land, which is located on the northern part of Hayes peninsula, on 13 April 2023. The black circles in (b) denote the sampling sites from St. 1 to St. 9, and the black line denotes dog sledge route. The gray contours in (b) are drawn from the Greenland Mapping Project 2 (GIMP-2) Digital Elevation Model version 2.

[Figure]

**Figure. 7 (in this file): Elevation above sea level of each station.** Gray values denote the distance from St. 1 to each station.

**Reviewer comment:**

In figure 5, in addition to changing colours between total and nss values, it would also be useful to change the symbols

**Author reply:**

Symbols will be removed from the figure corresponding to the vertical profile of $\delta^{18}O$ and ion species because we will plot the step-width graph. The example of figure corresponding to the vertical profile of $\delta^{18}O$ and ion species was shown in Fig. 2 and Fig. 3 in this file.

**Reference:**

Charlson, R. J., Lovelock, J. E., Andreae, M. O., and Warren, S. G.: Oceanic phytoplankton, atmospheric sulphur, cloud albedo and climate, Nature, 326, 655–661, https://doi.org/10.1038/326655a0, 1987.

Jaffrezo, J.-L., Davidson, C. I., Legrand, M., and Dibb, J. E.: Sulfate and MSA in the air and snow on the Greenland

Ice Sheet, J. Geophys. Res. Atom., 99, 1241–1253, https://doi.org/10.1029/93JD02913, 1994.

Hersbach, H., Bell, B., Berrisford, P., Hirahara, S., Horányi, A., Muñoz-Sabater, J., Nicolas, J., Peubey, C., Radu,

R., Schepers, D., Simmons, A., Soci, C., Abdalla, S., Abellan, X., Balsamo, G., Bechtold, P., Biavati, G.,

Bidlot, J., Bonavita, M., De Chiara, G., Dahlgren, P., Dee, D., Diamantakis, M., Dragani, R., Flemming, J.,

Forbes, R., Fuentes, M., Geer, A., Haimberger, L., Healy, S., Hogan, R. J., Hólm, E., Janisková, M., Keeley,

S., Laloyaux, P., Lopez, P., Lupu, C., Radnoti, G., De Rosnay, P., Rozum, I., Vamborg, F., Villaume, S., and

Thépaut, J.: The ERA5 global reanalysis, Quart J Royal Meteoro Soc, 146, 1999–2049, https://doi.org/10.1002/qj.3803, 2020.

Kurosaki, Y., Matoba, S., Iizuka, Y., Niwano, M., Tanikawa, T., Ando, T., Hori, A., Miyamoto, A., Fujita, S., and

Aoki, T.: Reconstruction of Sea Ice Concentration in Northern Baffin Bay Using Deuterium Excess in a

Coastal Ice Core From the Northwestern Greenland Ice Sheet, JGR Atmospheres, 125, e2019JD031668, https://doi.org/10.1029/2019JD031668, 2020.

Kurosaki, Y., Matoba, S., Iizuka, Y., Fujita, K., and Shimada, R.: Increased oceanic dimethyl sulfide emissions in areas of sea ice retreat inferred from a Greenland ice core, Commun Earth Environ, 3, 327, https://doi.org/10.1038/s43247-022-00661-w, 2022.

Muñoz-Sabater, J., Dutra, E., Agustí-Panareda, A., Albergel, C., Arduini, G., Balsamo, G., Boussetta, S., Choulga,

M., Harrigan, S., Hersbach, H., Martens, B., Miralles, D. G., Piles, M., Rodríguez-Fernández, N. J., Zsoter,

E., Buontempo, C., and Thépaut, J.-N.: ERA5-Land: a state-of-the-art global reanalysis dataset for land applications, Earth Syst. Sci. Data, 13, 4349–4383, https://doi.org/10.5194/essd-13-4349-2021, 2021.

Stein, A. F., Draxler, R. R., Rolph, G. D., Stunder, B. J. B., Cohen, M. D., and Ngan, F.: NOAA's HYSPLIT

Atmospheric Transport and Dispersion Modeling System, Bulletin of the American Meteorological Society,

96, 2059–2077, https://doi.org/10.1175/BAMS-D-14-00110.1, 2015.

---

## Author Comment (AC2)

**Dear Reviewer, #2**

**Thank you very much for your valuable comments. Our responses and the changes that we plan to make in the revised manuscript are explained below. We filled in reviewer comments in black, author replies in blue, the proposed changes to the revised manuscript in red.**

**Reviewer comment:**

First, data from the ST9 snow pit indicate evidence of summer surface snowmelt. Such melting processes hinder the preservation of proxy records and introduce uncertainty in age-dating. As discussed in the manuscript (lines 164–165), water isotope records tend to become smoothed, and ion concentrations are altered due to refreezing of meltwater. Therefore, the interpretation of vertical variations in proxy concentrations should account for these site-specific characteristics. Particularly in Section 3.2 ("Spatial and temporal variations in water isotopes and chemical species"), the interpretation of ST9 data should reflect the impact of summer melt on concentration variability.

**Author reply:**

As you have pointed out, the water stable isotopes were smoothed and ion concentrations were relocated due to meltwater refreezing. Therefore, the amplitude of the seasonal variations in water stable isotopes were smaller and ion concentrations showed high peaks in the ice layer. We have already described the impact of the melt water refreezing on the seasonality of water stable isotopes and ion concentrations in our manuscript as follows.
=====
**3.1 Snowpack dating and annual accumulation**
~~

**3.2 Spatial and temporal variations in $\delta^{18}O$ and chemical species**
~~
=====

**Reviewer comment:**

The seasonal classification such as spring–summer vs. autumn–winter should be used consistently, and the discussion of concentration variability should be supported by statistical criteria due to no clear variability of proxies. For example, it is recommended to define peaks using either values above the mean or above the mean plus one standard deviation.

**Author reply:**

In accordance with your comment, we will unify the description of seasonal classification as spring-summer and autumn-winter.

We will define positive (negative) peaks of each ion according to statistical criteria, which were values above (below) the mean plus (minus) one standard deviation.

**Reviewer comment:**

Second, additional evidence is required to substantiate some of the manuscript's interpretations. For example, to support the discussion on atmospheric transport, the inclusion of backward trajectory modeling (e.g., frequency maps and cluster analyses) is recommended as supplementary information to identify source regions and air mass pathways.

**Author reply:**

Thank you for your valuable suggestion.

This study suggested that the oceanic aerosols were transported from the area near the western side of Prudhoe

Land, located on the NOW, throughout the year by analyzing $\delta^{18}O$ and chemical species in the snowpack in western side of Prudhoe Land. To make this assumption more reliable, we performed the backward trajectory analysis and created a frequency map of the air mass transportation (Fig. 1 in this file). Majority of air mass arriving at western side of Prudhoe Land was transported from southern Greenland via northern Baffin Bay and eastern NOW. The existing probability in the eastern NOW was particularly high. The backward trajectory analysis supported our assumption, which the primary source of oceanic aerosols was NOW polynya throughout the year.

Figure 1 in this file will be added to the supplementary material, and relevant explanations will also be added in our revised manuscript.

[Figure]

**Figure 1 (in this file): Existence probability of an air mass occurring during the past during 7 days reaching the St. 9 in whole of year from 2019–2023.** (a) and (b) display Arctic area (> 60°N) and around northwestern Greenland, respectively. Black circles show the position of the St. 9.

**Reviewer comment:**

Lines 44–70: The necessity of studying past environmental changes in the NOW region is well presented. However, further explanation is needed on how the current study site differs from the nearby SIGMA-A site, especially in terms of meteorological conditions like prevailing wind directions.

**Author reply:**

We will add a description of how this study site differs from the nearby SIGMA-A site, as follows.

=====

In the Prudhoe Land, a gentle valley separates the western region from the eastern region, where the SIGMA-A site is situated. (Fig. 1b). This valley could serve as a pathway for air masses transported from inland of the Greenland Ice Sheet to descend toward the coastal region, and these air masses are likely not transported to the western side of Prudhoe Land. If that is the care, the snowpack on the western part of Prudhoe Land could contain aerosols originating from the NOW without being mixed with aerosols from the interior of the Greenland Ice Sheet.
=====

**Reviewer comment:**

Line 68–69: Rephrase for clarity.

**Author reply:**

We have revised the text you kindly pointed out as follows.
=====

Water vapor and sea salt on the glacier facing the ocean were transported from the coast. On the other hand, water vapor, mineral dust, anthropogenic substances, and MSA on the eastern side of Prudhoe Land were transported from the central-west coast of Greenland, which is located around Disko Bay, via the central part of the Greenland Ice Sheet (Matoba et al., 2014).
=====

**Reviewer comment:**

Lines 104–105: Add information in the Supplementary Information regarding the design and cleaning procedures of the pre-cleaned stainless-steel tools used for snow pit sampling. Clarify the cleanliness specification of the Whirl-Pak polyethylene bags (e.g., part number, manufacturer).

**Author reply:**

We removed the oil contamination on the precleaned materials and tools using ethanol, and performed then ultrasonic cleaning in ultrapure water. The [®]Whirl-Pak polyethylene bags were produced by Nasco. We will add this description to our revised manuscript.

**Reviewer comment:**

Because sample depth resolution varies (2 cm, 3 cm, 5–10 cm), figures such as Figure 4 should adopt a step-wise format for clarity, not dot and line format.

**Author reply:**

In accordance with your comment, we have changed figure plots of water stable isotopes and ion concentrations to step-wise format (ex. Fig. 2 in this file).

[Figure]

**Figure 2 (in this file): Vertical profile of NH$_4^+$ at (a) St. 3 and (b) St. 9.** Green lines denote mean NH$_4^+$ across all observation depths. Orange and brown lines denote the mean NH$_4^+$ plus and minus one standard deviation across all observation depths, respectively. The LOD of NH$_4^+$ was < 0.0055 μeq L$^{-1}$.

**Reviewer comment:**

Line 112: Provide details about possible contamination during snow sample melting and bottling. if possible, field blank should be provided.

**Author reply:**

We did not make a field brank in this observation. The field brank from our previous research (Kurosaki et al., 2020), which was made following the same procedure as used in this observation, was below the detection limit in the ion chromatography analysis.

**Reviewer comment:**

Line 117: Include specifications of the analytical column (e.g., length, diameter), model/manufacturer of standard materials, and detection limits for each ion.

**Author reply:**

As you have commented out, we will include the details of analytical column, standard materials, and detections limits as follows.

=====

For the cations, separation was carried out with a Dionex CG-12 (4 × 50 mm) guard column, followed by a Dionex CS12-A (4 × 250 mm) separation column. Injection volume of samples was 500 μL. MSA (20 mM) was used as eluent, and flow-rate was kept 1.0 mL min-1. Dionex CDRS600 dynamically regenerated suppressor was used for conductivity suppression before conductivity cell. For the anions, separation was obtained with a Dionex AG-18 (4 × 50 mm) guard column and Dionex AS-18 (4 × 250 mm) separation column. Injection volume of samples was 1000 μL. KOH (23 mM) was used as eluent, and flow-rate was kept 1.0 mL min$^{-1}$. Dionex ADRS 600 dynamically regenerated suppressor was used for conductivity suppression before conductivity cell. 5-point calibration curves were used for quantitative determination of each ion. The 5-point calibration curves were constructed using standard solution (Fujifilm Waco) adjusted to 20, 50, 100, and 200 ppb with ultra-pure water. If the ion concentration of samples were outside the calibration range (> 200 ppb), it was remeasured using 500, 1000, 2000~3000, and 6000 ppb standard for the anions and 500, 1000, 2000, and 4000 ppb standard for the cations. Blanks were always evaluated before the calibration procedure. The analytical precision of the ion chromatography was < 5 % (at the measurement of 20 ppb standard). The limit of detection (LOD) was < 0.1 ppb. The limit of quantification (LOQ) was < 0.5 ppb.

=====

**Reviewer comment:**

Line 122: Specify the standard material used for stable water isotope analysis.

**Author reply:**

We will add the standard materials used for stable water isotope analysis in the method section as follows.

=====

We used the ultrapure water ($\delta^{18}O = -11.583$ and $\delta D = -77.2$), Antarctic iceberg ($\delta^{18}O = -20.4$ and $\delta D = -158.7$), snowpack on the Antarctic ice Sheet ($\delta^{18}O = -46.694$ and $\delta D = -370.7$) for calibration.

=====

**Reviewer comment:**

Line 139: Present snow density alongside depth.

**Author reply:**

We will add the vertical snow density in our supplementary materials (Fig. 3 in this file).

[Figure]

**Figure 3 (in this file): Vertical profiles of the snow density at (a) St.3 and (b) St.9.**

**Reviewer comment:**

Lines 143–144: Ice layers below 0.96 m in the ST9 snowpack suggest summer melting, which may affect proxy preservation. This is appropriately and kindly described in lines 163–168.

**Author reply:**

As you have pointed out, we also think that the ion concentrations showed high values in the ice layers due to meltwater refreezing. Therefore, we attributed the peaks at the ice layers to meltwater refreezing and the other peaks to the deposition of atmospheric aerosols. We have already described this sentence in our manuscript.

**Reviewer comment:**

Line 172: Calculate annual accumulation rates using snow density for each depth interval and present average values.

**Author reply:**

As you have pointed out, we calculated annual accumulation rates using snow density for each depth interval.

We will revise the text regarding the annual accumulation rates as follows.

=====

According to the snowpack dating approach described above, the snow densities, which were the average of the bulk density of ice core at St. 9 (Fig. S3), from autumn–winter 2020 to autumn–winter 2021 and from autumn–

winter 2019 to autumn–winter 2020 were 477 kg m$^{-3}$ and 497 kg m$^{-3}$, respectively. Using these mean densities, the annual accumulations were calculated to be 0.41 m w.eq. yr$^{-1}$ and 0.56 m w.eq. yr$^{-1}$, respectively.

=====

**Reviewer comment:**

Line 183: Indicate the MSA detection limit as a line in Figure 4. Clarify dating below 3.4 m at ST9 (the conclusion mentions dating down to 4.5 m).

**Author reply:**

We have included the limit of detections (LOD) for each ion in captions of related figures because they were too small to be clearly shown in the figures.

We will edit the captions as follows:

[Figure]

**Figure 5 (in this file): Vertical profile of $NH_4^+$ at (a) St. 3 and (b) St. 9.** Green lines denote mean $NH_4^+$ across all observation depths. Orange and brown lines denote the mean $NH_4^+$ plus and minus one standard deviation across all observation depths, respectively. The LOD of $NH_4^+$ was $< 0.0055$ µeq $L^{-1}$.

**Reviewer comment:**

Line 188: Include NO3- data.

**Author reply:**

We have deleted the relevant sentence, following the suggestion of another reviewer.

**Reviewer comment:**

Line 196: Use nss-Ca2+ to interpret dust transport. Since nss-K+ and nss-Mg2+ mostly show negative values, suggesting major marine influence, omit these from discussion and Table 1.

**Author reply:**

As you have commented out, we have removed the nss-Mg2+ and nss-K+ in our discussion and Table 1.

**Reviewer comment:**

Lines 197–209: Explain shortly the notable difference in $\delta^{18}O$ between the upper layer (0–0.7 m) and the deeper layer.

**Author reply:**

The snow stratigraphy from 0.0 to 0.96 m at St. 9 were the rounded grains, faceted crystals, or depth hoar, whereas the melt forms prevailed below 0.96 m. The $\delta^{18}O$ in the snowpack below 0.96 m were smoothed by melting, thereby seasonal variations in $\delta^{18}O$ were smaller than the upper layer.

We have already described this sentence in our manuscript as follows.

=====

Snowpack below 0.96 m corresponded to previous summer and before it because the amplitude of seasonal variation of $\delta^{18}O$ and d-excess below 0.96 m were smaller than those in the shallower layers from 0.00–0.96 m because of summer melting.

=====

**Reviewer comment:**

Line 201: Present backward trajectory modeling results to support atmospheric transport path interpretations.

**Author reply:**

In accordance with your comment, we will add backward trajectory analysis (Fig. 1 in this file). We suggested that the south-to-north gradient of the $\delta^{18}O$ results from water vapor from the southern coast to northern inland area by the southerly winds. The 7-days backward trajectory of air mass arriving at St. 9 supported this suggestion, showing that the majority of air mass was transported from northern Baffin Bay and eastern NOW.

We will add the description regarding the backward trajectory analysis as follows.

=====

**2.3 Backward trajectory**

To investigate the source region and transport pathway of water vapor and aerosols contained in ice core at the St. 9 site, we analysed air mass position along the backward trajectory from the St. 9 site during the past 7 days using the National Oceanographic and Atmospheric Administration (NOAA) Hybrid Single-Particle Lagrangian Integrated Trajectory (HYSPLIT) model (Stein et al., 2015) and National Centers for Environmental Prediction (NCEP) reanalysis data. The initial positions of air mass were set at 50, 500, 1000, 1500 m above ground level over the St. 9 site. The initial date and time were every 6 h from 2019 to 2023. We calculated the probability of the existence of an air mass with a 1° resolution. Considering the water vapor and aerosols supply from the ocean and land surface, we excluded air mass over 1000 m above ground level. The existence probability was weighted by the daily amount of precipitation when the air mass arrived at the St. 9 site. The daily amount of precipitation was extracted from the ERA5 reanalysis dataset (Hersbach et al., 2020).

**3.2.1 $\delta^{18}O$**

The spatial variations in $\delta^{18}O$ in the surface snow showed maximum and minimum values at St. 3 (−19.12 ‰) and St. 9 (−37.21 ‰), respectively (Fig. S4a). The average $\delta^{18}O$ value from 0.00 to 1.01 m at St. 3 was greater than that at St. 9 (St. 3: −22.03 ‰; St. 9: −29.12 ‰) (Table 1). The $\delta^{18}O$ values in surface snow and the snowpack decreased from the seacoast toward the inland site. The past 7 days backward trajectory arriving at the St. 9 also exhibited that majority of air mass was transported from the south of the St. 9, situated on the northern Baffin Bay and eastern NOW (Fig. S5). We suggest that the south-to-north gradient of $\delta^{18}O$ results from water vapor, which originates from northern Baffin Bay and eastern NOW, transport from the southern coast to the northern inland area by southerly winds.
=====

**Reviewer comment:**
Line 216: Interpretation in Figure 6c should align with the seasonal framework in Figure 6b.

**Author reply:**
We will correct "summer to winter" to "spring-summer to autumn-winter" according to the seasonal framework of the dating in the snowpack at St. 9.

We will revise the text that you have pointed out as below.
=====
The difference in $\delta^{18}O$ values between St. 3 and St. 9 increased from spring–summer to autumn–winter from 2022–2023 and decreased until spring in 2023 (Fig. 3c).
=====

**Reviewer comment:**
Lines 222, 232: Revise for clarity.

**Author reply:**
We will revise the text that you have pointed out as below.
=====
Therefore, the difference of the $Cl^-/Na^+$ ratio in snowpack from sea-water ratio often reveals sea salt modification within the atmosphere.
=====

**Reviewer comment:**

Line 239: Provide supporting data

**Author reply:**

We will add the 7-days backward trajectory analysis (Fig. 1 in this file). We suggested the sea salt observed in this study could be transported along a short distance pathway without reactions with $H_2SO_4$ and $HNO_3$ during transportation. The backward trajectory analysis supported this suggestion, showing that the air mass frequency passed over the eastern NOW ocean, located near the St. 9, was high throughout the year.

**Reviewer comment:**

Line 278: Explain nitrate concentration increases due to melting/refreezing. if possible, explain shortly or provide references. Revise "positive peaks" to just "peaks."

**Author reply:**

The $NO_3^-$ tend to move easily with meltwater and become concentrated during refreezing (Matoba et al., 2002). We will add this text to our manuscript.

We will correct the "positive peaks" to "peaks" in our revised manuscript.

**Reviewer comment:**

Table 1: Replace nss-$K^+$ with $K^+$ and nss-$Mg^{2+}$ with $Mg^{2+}$ data.

**Author reply:**

In accordance with your comment, we have replaced nss$K^+$ with $K^+$ and nss$Mg^{2+}$ with $Mg^{2+}$ in Table 1 (in this file).

**Table 1 (in this file). Mean values and standard deviations of several ion species and water stable isotopes between St. 3 and St. 9.** The mean values at St.9 were obtained at depths from 0.00 to 1.01 m.

| | $Na^+$ (µeq L$^{-1}$) | $NH_4^+$ (µeq L$^{-1}$) | $K^+$ (µeq L$^{-1}$) | $Mg^{2+}$ (µeq L$^{-1}$) | $nssCa^{2+}$ (µeq L$^{-1}$) | MSA (µeq L$^{-1}$) | $Cl^-$ (µeq L$^{-1}$) | $nssSO_4^{2-}$ (µeq L$^{-1}$) | $NO_3^-$ (µeq L$^{-1}$) | $Cl^-/Na^+$ (µeq L$^{-1}$) | $\delta^{18}O$ (‰) | $\delta D$ (‰) | d-excess (‰) |
|---|---|---|---|---|---|---|---|---|---|---|---|---|---|
| mean(St.3) | 133.70 | 0.61 | 2.77 | 22.69 | -0.46 | 0.01 | 155.53 | 1.51 | 1.32 | 1.19 | -22.03 | -160.27 | 15.93 |
| std. (St.3) | 181.51 | 0.45 | 4.26 | 27.51 | 0.87 | 0.02 | 206.34 | 3.45 | 1.19 | 0.15 | 2.22 | 17.58 | 2.12 |
| mean(St.9) | 34.17 | 0.04 | 0.68 | 7.11 | 0.85 | 0.01 | 41.37 | 0.40 | 0.60 | 1.17 | -29.12 | -218.69 | 14.29 |
| std. (St.9) | 54.12 | 0.22 | 1.25 | 9.12 | 3.67 | 0.01 | 65.33 | 0.92 | 0.44 | 0.12 | 6.46 | 52.27 | 2.66 |

**Reviewer comment:**

Avoid repeating earlier content. Summarize only the most significant findings and implications.

**Author reply:**

We will summarize the conclusion section as follows, and we will avoid some repetitions such as "western side of Prudhoe Land".

=====

**4 Conclusion**

We conducted glaciological observations from 9−11 April 2023 on the western side of Prudhoe Land in northwestern Greenland facing the NOW to elucidate the source conditions and transportation processes of water vapor and aerosols in this region. The dating of the snowpack at St. 9, which is located at the inland of the western side of Prudhoe Land, revealed that the layer at a depth of 4.20 m corresponded to 3.5 years. The average annual accumulation at St. 9 was 0.49 m w.eq. $yr^{-1}$.

The snowpacks on the western side of Prudhoe Land contained aerosols from distant sources, such as remote dust and anthropogenic aerosols, in early spring–summer layers. On the other hand, they also contained aerosols from local sources such as ocean biological activity and frost flowers in the NOW and local dust around the coast of northwestern Greenland during other seasons, unlike the inland of the Greenland Ice Sheet. Moreover, we noted that the snowpacks were able to trace the poleward heat and moisture transport event along Baffin Bay during winter.

Arctic climate warming caused decreases in the sea ice thickness and concentration over the last few decades in the NOW and could influence clouds and precipitation following changes in sea ice and biological activities in the NOW. We found for the first time that the environmental changes in the NOW can be elucidated by the snowpack and ice core on the western side of the Prudhoe Land. We suggest that the chemical substances in the deeper ice core from this region could help explain the multidecadal variations in the sea ice, biological activities, and related water and aerosol circulation around the NOW and could develop to understand the accurate future projections of environmental change in this region.

=====

**References:**

Kurosaki, Y., Matoba, S., Iizuka, Y., Niwano, M., Tanikawa, T., Ando, T., Hori, A., Miyamoto, A., Fujita, S., and Aoki, T.: Reconstruction of Sea Ice Concentration in Northern Baffin Bay Using Deuterium Excess in a Coastal Ice Core From the Northwestern Greenland Ice Sheet, JGR Atmospheres, 125, e2019JD031668, https://doi.org/10.1029/2019JD031668, 2020.

Matoba, S., Narita, H., Motoyama, H., Kamiyama, K., and Watanabe, O.: Ice core chemistry of Vestfonna Ice Cap in Svalbard, Norway, J.‐Geophys.‐Res., 107, https://doi.org/10.1029/2002JD002205, 2002b.

Matoba, S., Yamasaki, T., Miyahara, M., and Motoyama, H.: Spatial variations of δ18O and ion species in the snowpack of the northwestern Greenland ice sheet, Bulletin of Glacier Research, 32, 79–84, https://doi.org/10.5331/bgr.32.79, 2014.

---

## Author Response (AR1)

- 1 Dear Dr. Krystyna Kozioł,
- 2 Thank you very much for your kind comments. We revised our manuscript following your comments. We
- 3 filled in your comments in the black and author replies in the blue. All line numbers of this manuscript have
- 4 linked to our revised manuscript. In the revised manuscript, edits made based on your comments are
- 5 highlighted in green.

**Editor comment:**

- 8 1. In lines 61-62 of Response to Reviewer#1, the Authors mention: "We removed the oil contamination on the
- 9 precleaned materials and tools using ethanol, and performed then ultrasonic cleaning in ultrapure water." It is
- unclear to me why there would be any oil contamination involved. Did you mean potential contamination with
- organic compounds in general?

12 13

**Author reply:**

- 14 As you have pointed out, we used ethanol to remove the potential contamination by organic compounds in general,
- not specifically oil, on materials and tools used for contamination removal. We have revised this sentence at line
- 16 116–117 in our revised manuscript.

17 18

**Editor comment:**

- 2. Was there a certified reference material used to ensure the quality of the ion chromatography analyses? If yes,
- 20 please describe; if not, please justify.

2122

**Author reply:**

- We used standard solutions for ion chromatography produced by FUJIFILM Wako Pure Chemical corporation for
- 24 the absolute calibration determination by ion chromatography. We have added this description at line 130–132 in
- 25 our revised manuscript.

2627

**Editor comment:**

- 3. In lines 144-145 of Response to Reviewer#1, the Authors mention: "The snowpack from 0.72 m-1.15 m
- 29 corresponded to spring to summer in 2022 from existence of ice layer" such phrasing is awkward, I would assume
- that it was deduced from the existence of ice layer that the snowpack layer corresponded to a certain time.

31

**32 **Author reply:**

- We have corrected the phrase you have pointed out as follows:
- 34 "The snowpack from 0.72 m-1.15 m was interpreted to correspond to the spring-summer period in 2022, due to
- 35 the presence of ice layer, high  $\delta^{18}$ O value and high MSA concentrations (Fig. 2 and Fig. S2)."
- This revised phrase has been added at line 197–199 in our revised manuscript.

3738

**Editor comment:**

- 4. In the suggested revised Section 2.3., there occur phrases such as "probability of existence", "existence
- 40 probability", "existing probability" what do they mean? Do they refer to the probability of air mass inflow from
- a certain direction or sector?

- 43 **Author reply:**
- 44 The existence probability was defined as the proportion of backward trajectories originating from the St. 9 that
- passed through each 1°×1°grid cell. The calculation procedure was as follows:
- 46 First, we counted the number of times the backward trajectories originating from St. 9 passed through each grid
- 47 cell. Second, we normalized the count for each grid cell by dividing it by the total number of the trajectory passes
- 48 across all grid cells.
- We have added this description at line 168–171 in our revised manuscript.

5051

**Editor comment:**

- 52 5. In lines 289-290 and 301-302 of Response to Reviewer#1, the Authors mention: "The mean temperature
- differences in autumn and winter were more negative than that in summer." and "that the altitude gradient of surface
- 54 air temperature in the western side of Prudhoe Land was steeper in winter than in summer.". The "more negative"
- 55 phrasing is confusing, since the difference between temperatures is an absolute value and therefore cannot be
- 56 negative. However, the proposed sentence for the manuscript reads correct if the difference was higher in winter.
- Please double-check that this was the intended meaning.

58

59

**Author reply:**

- As you have pointed out, the original phrasing was ambiguous and may have caused confusion. We have corrected
- the unit of temperature from degrees Celsius to Kelvin. With that in mind, we calculated the temperature difference
- between St. 3 and St. 9. The temperature difference was smallest in summer and increased toward winter (Fig. S6).
- The mean temperature differences in autumn and winter were larger than that in summer (Fig. S7). We have revised
- this description at line 249–253 in our revised manuscript.

65 66

**Editor comment:**

6. There is also a typo in line 80 of Response to Reviewer#2 ("care" instead of "case").

68

67

**69 **Author reply:**

We have corrected the word "If that is the care" to "If that is the case".

**72 Figures:**

Figure S6: Diurnal variations in (a) 2 m air temperature at St. 3 and St. 9, and (b) 2 m air temperature difference between St. 3 and St. 9.

Figure S7: Seasonal variations in the difference of 2 m air temperature between St. 3 and St. 9.

- 86 Dear Reviewer, #1
- 87 Thank you very much for your valuable comments. We revised our manuscript following your comments.
- We filled in reviewer comments in the black and author replies in the blue. All line numbers of this
- 89 manuscript have linked to our revised manuscript. In the revised manuscript, edits made based on reviewer
- 90 comments are highlighted in yellow.

- **Reviewer comment:**
- 93 In this paper entitled "Characteristics of snowpack chemistry on the coastal region in the northwestern Greenland
- 94 Ice Sheet facing the North Water", authors present an interesting observation of the effect that polynia North water
- 95 (NOW) has on aerosol circulation and precipitation. the results are obtained from measurements of major ions,
- MSA and water isotopic analyses at 9 surface snow sampling sites, 2 snow-pit sites and 1 ice core. The text is well
- 97 structured a detailed introduction, however the drafting in general should be improved as there are numerous
- 98 repetitions and in some parts the reading is difficult to understand. In particular, the section 3.2 has to be improved.
- 99 The conclusions have to be focused on the main goals obtained in this paper. It is very long and I suggest to
- summarize, avoiding to repeat the results and discussion.

101

- 102 **Author reply:**
- To improve the overall logical flow and readability in section 3.2, we have added individual sub-sections for  $\delta^{18}$ O
- and ion species. Figures of  $\delta^{18}$ O and each ion concentration were presented separately within their respective sub-
- sections.
- We also revised the Conclusion section by summarizing its content and removing some repetitive statements.

107

- 108 **Reviewer comment:**
- On lines 105-106: "The snow sampling intervals at St. 3 were 0.02 m from 0.00 to 0.20 m and 0.03 m from 0.20 to
- 1.01 m, and the snow sampling intervals at St. 9 were 0.02 m from 0.00 to 0.20 m and 0.03 m from 0.30 to 1.08
- m." Why was the sampling interval changed?

112

- 113 **Author reply:**
- 114 If we could sample the entire snowpack at short intervals, we would have been able to discuss the temporal
- variations in chemical components with short time intervals. However, we changed sampling interval partway
- through the snowpack, because we had limitations on the number of snow samples that could be transported by
- 117 dog sledges.

118119

- **Reviewer comment:**
- Lines 104, 108, 110. The authors told of precleaned materials and tools, but the cleaning procedure is not described.

121

- 122 **Author reply:**
- The cleaning procedure has been added to the Method section at line 116–117.

**125 **Reviewer comment:**

On line 107: Why was the ice core only sampled at one site? could be used for comparison at least with st3. 126

127

**128 **Author reply:**

We prioritized sampling as much as possible at St. 9 because of the limitation on the number of snow samples that 129

could be transported by dog sledges.

130 131

132

**Reviewer comment:**

Line 115: "methane sulfonate— (hereafter referred to as MSA)" already defined in the introduction

133 134 135

**Author reply:**

The definition of MSA moved at line 51–52.

137 138

136

**Reviewer comment:**

- 139 Lines 116-118. Please add several details about the analytical methods or some references. In particular, the authors
- declared only the columns used for cations and anions without any specific important details such as dimensions. 140
- Other important details are flows, injection volumes, instruments used, suppressors, detectors. No specific details 141
  - about the quantification methods are reported. I suppose that you used external calibration curves, but which are
- 143 the linear ranges, and which are the RCM used for quantification. In summary, please improve the method and
- 144 quality control section about the ionic analysis.

145 146

142

**Author reply:**

- 147 For the cations, separation was carried out with a Dionex CG-12 (4 × 50 mm) guard column, followed by a Dionex
- CS12-A (4 × 250 mm) separation column. Injection volume of samples was 500 µL. MSA (20 mM) was used as 148
- 149 eluent, and flow-rate was kept 1.0 mL min-1. Dionex CDRS600 dynamically regenerated suppressor was used for
- conductivity suppression before conductivity cell. For the anions, separation was obtained with a Dionex AG-18 150
- 151 (4 × 50 mm) guard column and Dionex AS-18 (4 × 250 mm) separation column. Injection volume of samples was
- 152 1000 μL. KOH (23 mM) was used as eluent, and flow-rate was kept 1.0 mL min-1. Dionex ADRS 600 dynamically
- regenerated suppressor was used for conductivity suppression before conductivity cell. The absolute calibration 153
- curve method was used for quantitative determination of each ion concentration. For the absolute calibration 154
- determination by ion chromatography, we used the standard solutions for ion chromatography produced by
- FUJIFILM Wako Pure Chemical corporation, diluted to 20, 50, 100, and 200 ppb with ultra-pure water. If the ion 156
- 157 concentration of samples were outside the calibration range (> 200 ppb), it was remeasured using 500, 1000, 2000–
- 3000, and 6000 ppb standard for the anions and 500, 1000, 2000, and 4000 ppb standard for the cations. Blanks 158
- 159 were always evaluated before the calibration procedure.
- 160 We have added this text regarding the analytical method of the ion chromatography at line 123–134.

161

162

155

**Reviewer comment:**

- Lines 117-119: Has the ion chromatography method used been validated in previous works? If yes, indicate them,
- if not, insert a section on validation.

- 166 **Author reply:**
- 167 The ion chromatography method had been validated by the previous work (Kurosaki et al., 2020; Kurosaki et al.,
- 168 2022). We have added this description to the method section at line 136–137.

169

- 170 **Reviewer comment:**
- Lines 119-120: "The samples exhibiting large peak were measured multiple times, to confirm that any large peak
- in ion concentration was not caused by analytical errors." What is meant?

173

- 174 **Author reply:**
- 175 If the ion concentration of samples were outside the calibration range (> 200 ppb), it was remeasured using 500,
- 176 1000, 2000~3000, and 6000 ppb standard for the anions and 500, 1000, 2000, and 4000 ppb standard for the cations.
- 177 Blanks were always evaluated before the calibration procedure.
- 178 We have added this description at line 132–134.

179

- 180 **Reviewer comment:**
- Lines 156-165 Text is not clear

182

- 183 **Author reply:**
- We have revised the text you kindly pointed out at line 194–204.

185

- 186 **Reviewer comment:**
- Section 3.2. Following stratigraphic analysis and evaluation of snowpack density, it may be more informative to
- express data in terms of fluxes rather than concentrations, so in the subsequent data analysis one could avoid
- distinguishing peaks attributed to atmospheric deposition from those of melting and refreezing

190

- 191 **Author reply:**
- 192 As you have pointed out, the deposition flux is sometimes more suitable when discussing the deposition amount of
- atmospheric aerosols for quantitatively. However, we cannot discuss the deposition flux because we did not collect
- the snow density with high resolution along the snow depth. Therefore, we qualitatively discussed the seasonal
- characteristics of ion species based on their concentration.

196 197

- **Reviewer comment:**
- Lines 188–190: Introducing all figures at the beginning of the section may lead to confusion. Since the discussion
- begins with Fig. 5, it would be more effective to present the figures sequentially, in alignment with the narrative.

200

201 **Author reply:**

In accordance with your comment, we have revised the order of figures. We have presented the  $\delta^{18}O$  and each ion 202 concentration within the relevant sub-sections of section 3.2, displaying the figures sequentially. 203 204 205 **Reviewer comment:** Line 194: "We applied the concentration unit as  $\mu$ eq L—" Information that is already made explicit in the following 206 207 graphs 208 209 **Author reply:** This sentence notes that " $\mu$  eq L-1" was used as the unit of ion concentration in equation (1). The editor requested 210 that the concentration unit should state clearly for the equation (1). 211 212 213 **Reviewer comment:** Line 201: "We suggest that the spatial variation in the  $\delta^{18}$ O results from water vapor transport from the southern 214 coast to the northern inland area by southerly winds." Might it be useful to indicate figure 9 by referring to the 215 direction of the prevailing winds? 216 217 **Author reply:** 218 Thank you for your comment. We suggested that the south-to-north gradient of the  $\delta^{18}$ O results from water vapor 219 from the southern coast to northern inland area by the southerly winds. We have performed the backward trajectory 220

analysis and analyzed the probability map of air mass transportation to make this assumption more reliable (Fig. 1

in this file). The 7-days backward trajectory of air mass arriving at St. 9 showed that the majority of air mass was

We have added this description at line 226-229. We have also added the method of backward trajectory at line

transported from the south of St. 9, situated on northern Baffin Bay and eastern NOW.

162–174. Figure 1 in this file has been added to the supplementary materials.

221

222

223

Figure 1 (in this file): Existence probability of an air mass occurring during the past during 7 days reaching the St. 9 in whole of year from 2019–2023. (a) and (b) display Arctic area (> 60°N) and around northwestern Greenland, respectively. Black circles show the position of the St. 9.

**Reviewer comment:**

Line 210. Please add "(figure 6)" to help readers or start the sentence introducing the Figure 6 and its meaning.

**Author reply:**

226

227

228229

230231232

233234

235

236

237238

We have added an explanation of the relevant figure at the beginning of the paragraph discussing the vertical profiles of  $\delta^{18}$ O at St. 3 and St. 9 (lines 238–239).

**Reviewer comment:**

Figure 6: I suggest using the season and year instead of Roman numerals, as this would facilitate interpretation. This recommendation may also apply to the other figures. It is somewhat difficult to follow the discussion, as it requires frequently switching between different figures.

**Author reply:**

To improve the overall logical flow and readability in section 3.2, we have added individual sub-sections for  $\delta^{18}O$  and ion species. Figures of  $\delta^{18}O$  and each ion concentration have been presented separately within their respective sub-sections. The seasonal divisions in each figure have been revised from Roman numerals to explicit labels indicating the season and year (ex. Fig. 2 in this file).

Figure 2 (in this file): Vertical profile of  $NH_4^+$  at (a) St. 3 and (b) St. 9. Green lines denote mean  $NH_4^+$  across all observation depths. Orange and brown lines denote the mean  $NH_4^+$  plus and minus one standard deviation across all observation depths, respectively. The LOD of  $NH_4^+$  was

Figure 3 (in this file): Vertical profile of  $\delta^{18}$ O. (a) and (b) show  $\delta^{18}$ O values at St. 3 and St.9, respectively. (c) shows difference between St.3 and St.9 in terms of  $\delta^{18}$ O. i–vii denote seasons from 2019 to 2023. i, iii, v, and vii denote from autumn to winter period from 2022–2023, 2021–2022, 2020–2021, and 2019–2020, respectively. ii, iv, and vi denote from spring to summer in 2022, 2021, and 2020, respectively.

**Reviewer comment:**

Line 217-218: "We suggest that the altitude gradient of the surface air temperature in winter was greater than that in summer in the western region of Prudhoe Land." could this statement also be confirmed using atmospheric models for specific sites?

**Author reply:**

We estimated the difference in surface air temperature between St. 3 and St. 9 using ERA5-Land reanalysis dataset (Fig. 4 and Fig. 5 in this file). The temperature difference was smallest in summer and increased toward winter. The mean temperature differences in autumn and winter were larger than those in summer. This result supports our suggestion, based on water stable isotope, that the altitude gradient of surface air temperature in the western side of Prudhoe Land was steeper in winter than in summer.

Figure 4 (in this file): Diurnal variations in (a) 2 m air temperature at St. 3 and St. 9, and (b) 2 m air temperature difference between St. 9 and St. 3.

Figure 5 (in this file): Seasonal variations in the difference of 2 m air temperature between St. 9 and St. 3.

**Reviewer comment:**

Lines 306-309: there are many repetitions of "the concentration of MSA". Same in the conclusions with "The snowpack on the western side of Prudhoe Land".

**Author reply:**

Thank you for your kind comment. We have revised the text that you have pointed out at line 403–406 and 471–486.

**Reviewer comment:**

General comment on the conclusions: from figure 1 sampling sites 1 to 5 (or 6) are in a valley. has this aspect been taken into consideration? could it have an impact on the final considerations?

**Author reply:**

I appreciate your valuable comment.

Because the topography in the western side of Prudhoe Land is smooth (Fig. 6 in this file) and the glacier is broad and relatively low gradient, we think that the enhancement of vertical convection and downslope wind caused by the valley topography are insignificant on the large-scale water vapor and aerosol circulation around the western side of the Prudhoe Land.

Figure 6 (in this file): Maps of the sampling sites. (a) shows location of the snowpit and ice core sampling sites in this study (St. 9) and previous studies (SIGMA-A, SIGMA-D, and NEEM) in the northwestern Greenland Ice Sheet. The dashed polygon in (a) denotes the approximate location of the NOW. Hayes peninsula in the northwestern Greenland is located between Kane Basin in the north and Melville Bay in the south. (b) shows Landsat-8 image around St. 9 and SIGMA-A of Prudhoe Land, which is located on the northern part of Hayes peninsula, on 13 April 2023. The black circles in (b) denote the sampling sites from St. 1 to St. 9, and the black line denotes dog sledge route. The gray contours in (b) are drawn from the Greenland Mapping Project 2 (GIMP-2) Digital Elevation Model version 2.

**Other comments:**

**Reviewer comment:**

- In figure 1b it might be useful to include a dimensional scale to give an idea of the distances.
- Similarly, in figure 2, in addition to the distance expressed in latitude, could a conversion to km be useful?

**Author reply:**

Thank you for your ideas. We have added the scale of distance and north arrow in Figure 1b (Fig. 6 in this file), and the distance from St. 1 to each sampling station in supplementary figure S1 (Fig. 7 in this file).

Figure. 7 (in this file): Elevation above sea level of each station. Gray values denote the distance from St. 1 to each station.

**Reviewer comment:**

In figure 5, in addition to changing colours between total and nss values, it would also be useful to change the symbols

**Author reply:**

Symbols have been removed from the figure depicting the vertical profiles of δ¹8O and ion species, as step-width graph were used in Figure 2–10.
References:
Kurosaki, Y., Matoba, S., Iizuka, Y., Niwano, M., Tanikawa, T., Ando, T., Hori, A., Miyamoto, A., Fujita, S., and Aoki, T.: Reconstruction of Sea Ice Concentration in Northern Baffin Bay Using Deuterium Excess in a Coastal Ice Core From the Northwestern Greenland Ice Sheet, JGR Atmospheres, 125, e2019JD031668, https://doi.org/10.1029/2019JD031668, 2020.
Kurosaki, Y., Matoba, S., Iizuka, Y., Fujita, K., and Shimada, R.: Increased oceanic dimethyl sulfide emissions in areas of sea ice retreat inferred from a Greenland ice core, Commun Earth Environ, 3, 327, https://doi.org/10.1038/s43247-022-00661-w, 2022.

- 361 Dear Reviewer, #2
- Thank you very much for your valuable comments. We revised our manuscript following your comments.
- We filled in reviewer comments in the black and author replies in the blue. All line numbers of this
- 364 manuscript have linked to our revised manuscript. In the revised manuscript, edits made based on reviewer
- 365 comments are highlighted in blue.

- **Reviewer comment:**
- First, data from the ST9 snow pit indicate evidence of summer surface snowmelt. Such melting processes hinder
- the preservation of proxy records and introduce uncertainty in age-dating. As discussed in the manuscript (lines
- 370 164–165), water isotope records tend to become smoothed, and ion concentrations are altered due to refreezing of
- meltwater. Therefore, the interpretation of vertical variations in proxy concentrations should account for these site-
- 372 specific characteristics. Particularly in Section 3.2 ("Spatial and temporal variations in water isotopes and chemical
- species"), the interpretation of ST9 data should reflect the impact of summer melt on concentration variability.

374375

- **Author reply:**
- 376 As you have pointed out, the water stable isotopes were smoothed and ion concentrations were relocated due to
- 377 meltwater refreezing. Therefore, the amplitude of the seasonal variations in water stable isotopes were smaller and
- ion concentrations showed high peaks in the ice layer. We have already described the impact of the melt water
- refreezing on the seasonality of water stable isotopes and ion concentrations at line 201–204, 305–306, 327–329,
- 380 354–357, and 378–380.

381 382

- **Reviewer comment:**
- The seasonal classification such as spring-summer vs. autumn-winter should be used consistently, and the
- discussion of concentration variability should be supported by statistical criteria due to no clear variability of
- proxies. For example, it is recommended to define peaks using either values above the mean or above the mean
- 386 plus one standard deviation.

387

- 388 **Author reply:**
- In accordance with your comment, we have unified the description of seasonal classification as spring-summer and
- 390 autumn-winter.
- We have defined positive (negative) peaks of each ion according to statistical criteria, which were values above
- 392 (below) the mean plus (minus) one standard deviation at line 303–305, 326–327, 353–354, 377–378, and 405–406.

393

394

- **Reviewer comment:**
- 395 Second, additional evidence is required to substantiate some of the manuscript's interpretations. For example, to
- support the discussion on atmospheric transport, the inclusion of backward trajectory modeling (e.g., frequency
- maps and cluster analyses) is recommended as supplementary information to identify source regions and air mass
- 398 pathways.

**Author reply:**

Thank you for your valuable suggestion.

This study suggested that the oceanic aerosols were transported from the area near the western side of Prudhoe Land, located on the NOW, throughout the year by analyzing  $\delta^{18}O$  and chemical species in the snowpack in western side of Prudhoe Land. To make this assumption more reliable, we performed the backward trajectory analysis and created a frequency map of the air mass transportation (Fig. 1 in this file). Majority of air mass arriving at western side of Prudhoe Land was transported from southern Greenland via northern Baffin Bay and eastern NOW. The existing probability in the eastern NOW was particularly high. The backward trajectory analysis supported our assumption, which the primary source of oceanic aerosols was NOW polynya throughout the year.

Figure 1 in this file has been added to the supplementary material, and relevant explanations have also been added at line 162–174 and 226–229 in our revised manuscript.

- 413 Figure 1 (in this file): Existence probability of an air mass occurring during the past during 7 days reaching
- 414 the St. 9 in whole of year from 2019–2023. (a) and (b) display Arctic area (> 60°N) and around northwestern
- Greenland, respectively. Black circles show the position of the St. 9.

417418

- **Reviewer comment:**
- Lines 44–70: The necessity of studying past environmental changes in the NOW region is well presented. However,
- 420 further explanation is needed on how the current study site differs from the nearby SIGMA-A site, especially in
- 421 terms of meteorological conditions like prevailing wind directions.

422

- 423 **Author reply:**
- In the Prudhoe Land, a gentle valley separates the western region from the eastern region, where the SIGMA-A
- site is situated. (Fig. 1b in our revised manuscript). This valley could serve as a pathway for air masses transported
- from inland of the Greenland Ice Sheet to descend toward the coastal region, and these air masses are likely not
- 427 transported to the western side of Prudhoe Land. If that is the case, the snowpack on the western part of Prudhoe
- 428 Land could contain aerosols originating from the NOW without being mixed with aerosols from the interior of the
- 429 Greenland Ice Sheet.
- We have added a description of how this study site differs from the nearby SIGMA-A site at line 83–87.

431

- 432 **Reviewer comment:**
- Line 68–69: Rephrase for clarity.

434

- 435 **Author reply:**
  - We have revised the text you kindly pointed out at line 66–69.

436437

- 438 **Reviewer comment:**
- Lines 104–105: Add information in the Supplementary Information regarding the design and cleaning procedures
- of the pre-cleaned stainless-steel tools used for snow pit sampling. Clarify the cleanliness specification of the Whirl-
- Pak polyethylene bags (e.g., part number, manufacturer).

442

- 443 **Author reply:**
- We removed the potential contamination by organic compounds in general on the material and tools used for
- contamination removal using ethanol, and then performed ultrasonic cleaning in ultrapure water. The ®Whirl-Pak
- polyethylene bags were produced by Nasco.
- We have added this description at line 110, 114, 116–117.

448

- **Reviewer comment:**
- Because sample depth resolution varies (2 cm, 3 cm, 5–10 cm), figures such as Figure 4 should adopt a step-wise
- 451 format for clarity, not dot and line format.

**453 **Author reply:**

- In accordance with your comment, we have changed figure plots of water stable isotopes and ion concentrations to
- step-wise format in Figures 2–10.

456

452

- 457 **Reviewer comment:**
- Line 112: Provide details about possible contamination during snow sample melting and bottling. if possible, field
- blank should be provided.

460

- 461 **Author reply:**
- We did not make a field brank in this observation. The field brank from our previous research (Kurosaki et al.,
- 463 2020), which was made following the same procedure as used in this observation, was below the detection limit in
- the ion chromatography analysis.

465466

**Reviewer comment:**

- Line 117: Include specifications of the analytical column (e.g., length, diameter), model/manufacturer of standard
- 468 materials, and detection limits for each ion.

469470

- **Author reply:**
- 471 For the cations, separation was carried out with a Dionex CG-12 (4 × 50 mm) guard column, followed by a Dionex
- 472 CS12-A (4 × 250 mm) separation column. Injection volume of samples was 500 μL. MSA (20 mM) was used as
- eluent, and flow-rate was kept 1.0 mL min-1. Dionex CDRS600 dynamically regenerated suppressor was used for
- 474 conductivity suppression before conductivity cell. For the anions, separation was obtained with a Dionex AG-18
- 475 (4 × 50 mm) guard column and Dionex AS-18 (4 × 250 mm) separation column. Injection volume of samples was
- 476 1000 μL. KOH (23 mM) was used as eluent, and flow-rate was kept 1.0 mL min-1. Dionex ADRS 600 dynamically
- 477 regenerated suppressor was used for conductivity suppression before conductivity cell. The absolute calibration
- 478 curve method was used for quantitative determination of each ion concentration. For the absolute calibration
- •
- determination by ion chromatography, we used the standard solutions for ion chromatography produced by
- FUJIFILM Wako Pure Chemical corporation, diluted to 20, 50, 100, and 200 ppb with ultra-pure water. If the ion
- concentration of samples were outside the calibration range (> 200 ppb), it was remeasured using 500, 1000, 2000–
- 482 3000, and 6000 ppb standard for the anions and 500, 1000, 2000, and 4000 ppb standard for the cations. Blanks
- 483 were always evaluated before the calibration procedure. The analytical precision of the ion chromatography was <
- 484 5 % (at the measurement of 20 ppb standard). The limit of detection (LOD) was < 0.1 ppb. The limit of
- quantification (LOQ) was < 0.5 ppb.
- We have included this text at line 123–136.

487 488

**Reviewer comment:**

Line 122: Specify the standard material used for stable water isotope analysis.

- 491 **Author reply:**
- We used the ultrapure water ( $\delta^{18}O = -11.583$  and  $\delta D = -77.2$ ), Antarctic iceberg ( $\delta^{18}O = -20.4$  and  $\delta D = -158.7$ ),
- snowpack on the Antarctic ice Sheet ( $\delta^{18}O = -46.694$  and  $\delta D = -370.7$ ) for calibration.
- We have added this description at line 146–148.

- **Reviewer comment:**
- 497 Line 139: Present snow density alongside depth.

498

- 499 **Author reply:**
- We have added the vertical snow density to our supplementary figure 3.

501

- **Reviewer comment:**
- Lines 143–144: Ice layers below 0.96 m in the ST9 snowpack suggest summer melting, which may affect proxy
- preservation. This is appropriately and kindly described in lines 163–168.

505

- 506 **Author reply:**
- As you have pointed out, we also think that the ion concentrations showed high values in the ice layers due to
- 508 meltwater refreezing. Therefore, we attributed the peaks at the ice layers to meltwater refreezing and the other
- 509 peaks to the deposition of atmospheric aerosols. We have already described this sentence at line 201–204, 305–
- 510 306, 327–329, 354–357, and 378–380.

511

- 512 **Reviewer comment:**
- 513 Line 172: Calculate annual accumulation rates using snow density for each depth interval and present average
- 514 values.

515

- 516 **Author reply:**
- As you have pointed out, we calculated annual accumulation rates using snow density for each depth interval.
- We have described the revised annual accumulation rates at line 208–211.

519

- **Reviewer comment:**
- Line 183: Indicate the MSA detection limit as a line in Figure 4. Clarify dating below 3.4 m at ST9 (the conclusion
- mentions dating down to 4.5 m).

523

- 524 **Author reply:**
- We have included the limit of detections (LOD) for each ion in captions of Figures 2 and 4–9 because they were
- too small to be clearly shown in the figures.

- **Reviewer comment:**
- 529 Line 188: Include NO3- data.

**530 **Author reply:** 531 We have deleted the relevant sentence, following the suggestion of another reviewer. 532 533 534 **Reviewer comment:** 535 Line 196: Use nss-Ca2+ to interpret dust transport. Since nss-K+ and nss-Mg2+ mostly show negative values, suggesting major marine influence, omit these from discussion and Table 1. 536 537 538 **Author reply:** As you have commented out, we have removed the nss-Mg2+ and nss-K+ in our discussion and Table 1. 539 540 **Reviewer comment:** 541 542 Lines 197–209: Explain shortly the notable difference in $\delta^{18}$ O between the upper layer (0–0.7 m) and the deeper layer. 543 544 545 **Author reply:** The snow stratigraphy from 0.0 to 0.96 m at St. 9 were the rounded grains, faceted crystals, or depth hoar, whereas 546 the melt forms prevailed below 0.96 m. The $\delta^{18}$ O in the snowpack below 0.96 m were smoothed by melting, thereby 547 seasonal variations in $\delta^{18}$ O were smaller than the upper layer. 548 We have described this sentence at line 181–184 and 199–201. 549 550 551 **Reviewer comment:** Line 201: Present backward trajectory modeling results to support atmospheric transport path interpretations. 552 553 554 **Author reply:** 555 In accordance with your comment, we have added backward trajectory analysis (Fig. 1 in this file). We suggested that the south-to-north gradient of the $\delta^{18}$ O results from water vapor from the southern coast to northern inland area 556 557 by the southerly winds. The 7-days backward trajectory of air mass arriving at St. 9 supported this suggestion, showing that the majority of air mass was transported from northern Baffin Bay and eastern NOW. 558 We have added the description regarding the backward trajectory analysis at line 162–174 and 226–229. 559 560 561 **Reviewer comment:** Line 216: Interpretation in Figure 6c should align with the seasonal framework in Figure 6b. 562 563 **Author reply:** 564 We have corrected "summer to winter" to "spring-summer to autumn-winter" according to the seasonal framework 565 of the dating in the snowpack at St. 9. 566**

We have revised the text that you have pointed out at line 245–246.

569 **Reviewer comment:** 570 Lines 222, 232: Revise for clarity. 571 572 **Author reply:** We have revised the text that you have pointed out at line 281–282. 573 574 **Reviewer comment:** 575 Line 239: Provide supporting data 576 577 **Author reply:** 578 579 We have added the 7-days backward trajectory analysis (Fig. 1 in this file). We suggested the sea salt observed in this study could be transported along a short distance pathway without reactions with H2SO4 and HNO3 during 580 transportation. The backward trajectory analysis supported this suggestion, showing that the air mass frequency 581 passed over the eastern NOW ocean, located near the St. 9, was high throughout the year. 582 We have added this description at line 287–289. 583 584 **Reviewer comment:** 585 Line 278: Explain nitrate concentration increases due to melting/refreezing, if possible, explain shortly or provide 586 references. Revise "positive peaks" to just "peaks." 587 588 **Author reply:** 589 The NO3- tend to move easily with meltwater and become concentrated during refreezing (Matoba et al., 2002). 590 We have added this text at line 356–357. 591 We have corrected the "positive peaks" to "peaks" at line 354. 592 593 594 **Reviewer comment:** Table 1: Replace nss-K+ with K+ and nss-Mg2+ with Mg2+ data. 595 596 **Author reply:** 597 In accordance with your comment, we have replaced nssK+ with K+ and nssMg2+ with Mg2+ in Table 1. 598 599 600 **Reviewer comment:** 601 Avoid repeating earlier content. Summarize only the most significant findings and implications. 602 **Author reply:** 603 604 We have summarized the conclusion section at line 471-486, and we have avoided some repetitions such as "western side of Prudhoe Land". 605 606

607

**References:**

| 608 | Kurosaki, Y., Matoba, S., Iizuka, Y., Niwano, M., Tanikawa, T., Ando, T., Hori, A., Miyamoto, A., Fujita, S., and |
|-----|-------------------------------------------------------------------------------------------------------------------|
| 609 | Aoki, T.: Reconstruction of Sea Ice Concentration in Northern Baffin Bay Using Deuterium Excess in a              |
| 610 | Coastal Ice Core From the Northwestern Greenland Ice Sheet, JGR Atmospheres, 125, e2019JD031668,                  |
| 611 | https://doi.org/10.1029/2019JD031668, 2020.                                                                       |
| 612 | Matoba, S., Narita, H., Motoyama, H., Kamiyama, K., and Watanabe, O.: Ice core chemistry of Vestfonna Ice Cap     |
| 613 | in Svalbard, Norway, J Geophys Res., 107, https://doi.org/10.1029/2002JD002205, 2002b.                            |
| 614 |                                                                                                                   |
| 615 |                                                                                                                   |

---

## Editor Decision (ED1)

**Review Comments**

Firstly, I would like to thank authors for their effort in responding to my coments.

At the moment, I found some sentences to be clarified in this article. In the section 3.3. intense poleward heat and moisture transport in winter from 2022-23

line 433: the phrase "leading to the coldest winter season" should be revised to "leading to the warmest winter season"

line 435: Figure 11a should be revised to Figure 11b

line 458: in the caption of the Figure 11, April 2022 to April 2023 should be corrected to June 2022 to May 2023. and also the sign convention should be revised to positive in Figure 9d, as southerly wind increases.

---

## Author Response (AR2)

**Dear Reviewer, #2**

Thank you very much for your valuable comments. We revised our manuscript following your comments. We filled in reviewer comments in the black and author replies in the blue. All line numbers of this manuscript have linked to our revised manuscript.

**Reviewer comment:**

line 433: the phrase "leading to the coldest winter season" should be revised to "leading to the warmest winter season"

**Author reply:**

We have revised the phrase "leading to the coldest winter season" to "leading to the warmest in winter from 2022–2023" at line 428.

**Reviewer comment:**

line 435: Figure 11a should be revised to Figure 11b

**Author reply:**

We have corrected the Figure number to "Fig. 11b" at line 430.

**Reviewer comment:**

line 458: in the caption of the Figure 11, April 2022 to April 2023 should be corrected to June 2022 to May 2023. and also the sign convention should be revised to positive in Figure 9d, as southerly wind increases.

**Author reply:**

We have corrected the period to "June 2022 to May 2023" at line 458 and revised the southerly wind to be represented as a positive value in Figure 11d.

Figure 11: Sea ice and meteorological conditions from June 2022 to May 2023 corresponding to snow depths ranging from 0.00–1.15 m at St. 9. (a) shows the sea ice concentration on the eastern side of NOW. (b) shows the temperature at 2 m a.g.l. in Siorapaluk (red line) and St. 9 (black line). (c) shows the southern component of the wind speed at 2 m a.g.l. in Siorapaluk (red line) and 10 m a.g.l. on the eastern side of the NOW (black line). (d) shows the meridional wind speed at 700 hPa along the eastern side of Baffin Bay. The values on the eastern side of the NOW were averaged over 75.0°–80.0°N and 75.0°–65.0°W. The values on the eastern side of Baffin Bay were averaged over 75.0°–65.0°W from 75.0° to 80.0°N, 75.0°–65.0°W from 75.0° to 80.0°N, 75.0°–65.0°W from 75.0° to 80.0°N. The positive and negative values of the meridional wind speed are the southern and northern parts of the wind, respectively. The black arrows denote the date from 5–6 December 2023.